# The role of mechano-regulated YAP/TAZ in erectile dysfunction

Mintao Ji[1,9], Dongsheng Chen [2,9], Yinyin Shu [1], Shuai Dong [1],
Zhisen Zhang [1], Haimeng Zheng[1], Xiaoni Jin[1], Lijun Zheng [1], Yang Liu [3],
Yifei Zheng[4,5], Wensheng Zhang[6], Shiyou Wang[2], Guangming Zhou[1],
Bingyan Li [7], Baohua Ji[4], Yong Yang [3] ✉, Yongde Xu [8] ✉ & Lei Chang [1] ✉

Phosphodiesterase type 5 inhibitors (PDE5is) constitute the primary therapeutic option for treating erectile dysfunction (ED). Nevertheless, a substantial proportion of patients, approximately 30%, do not respond to PDE5i treatment. Therefore, new treatment methods are needed. In this study, we identified a pathway that contributes to male erectile function. We show that mechano-regulated YAP/TAZ signaling in smooth muscle cells (SMCs) upregulates adrenomedullin transcription, which relaxed the SMCs to maintain erection. Using single-nucleus RNA sequencing, we investigated how penile erection stretches the SMCs, inducing YAP/TAZ activity. Subsequently, we demonstrate that YAP/TAZ plays a role in erectile function and penile rehabilitation, using genetic lesions and various animal models. This mechanism relies on direct transcriptional regulation of adrenomedullin by YAP/TAZ, which in turn modulates penile smooth muscle contraction. Importantly, conventional PDE5i, which targets NO-cGMP signaling, does not promote erectile function in YAP/TAZ-deficient ED model mice. In contrast, by activating the YAP/TAZ-adrenomedullin cascade, mechanostimulation improves erectile function in PDE5i nonrespondent ED model rats and mice. Furthermore, using clinical retrospective observational data, we found that mechanostimulation significantly promotes erectile function in patients irrespective of PDE5i use. Our studies lay the groundwork for exploring the mechano-YAP/TAZ-adrenomedullin axis as a potential target in the treatment of ED.

Erectile dysfunction (ED), also known as impotence, is characterized by the inability to maintain sufficient rigidity of the penile erection to accomplish copulation. An erection is accomplished by arterial dilation smooth muscle relaxation, and venous constriction around the penis. Consequently, the corpus cavernosum fills with and retains blood to achieve and maintain an erection[1,2]. The most commonly used medicines for ED are phosphodiesterase type 5 inhibitors (PDE5i), widely known by its commercial name Viagra[3]. PDE5is bind to the catalytic site of PDE5 to block the degradation of cGMP in smooth muscle cells (SMCs), thereby potentiating the effects of cGMP on smooth muscle

relaxation to prolong erections[4]. PDE5is, such as sildenafil and tadalafil, are the most common clinically used drugs to treat ED. However, approximately 30% of ED patients are classified as "nonresponders" to PDE5is[5], spurring interest in finding new treatment options.

During clinical practice, physiotherapies, such as vacuum erection devices (VEDs) and shock wave therapy (SWT), have been widely used to treat ED patients. VED relies on negative pressure to draw blood into the penis, and repeated application has been shown to improve ED[1,3,6,7]. SWT consists of noninvasive low-intensity sound waves that pass through erectile tissue, restoring natural erectile function[3,8]. However,

the underlying mechanism of these treatments remains to be elucidated.

Here, we have identified the activation of YAP/TAZ through the stretching of SMCs as a contributing factor in maintaining an erection. Nuclear YAP/TAZ directly regulates the transcription of adrenomedullin (ADM), a locally acting hormone controlling vascular tone. Diffusing ADM promotes smooth muscle relaxation to sustain an erection. Thus, our work elucidated a physiological step in penile erection and identified a mechano-regulated YAP/TAZ-ADM molecular axis as its underlying molecular mechanism.

## Results

### An erect penis mechanically stretches SMCs and activates YAP/TAZ

Healthy men experience nocturnal penile tumescence (NPT), which is the spontaneous erection of the penis during sleep, and its regular occurrence of three to five times a night is a widely known clinical hallmark of erectile function[9]. Men with the onset of ED lose NPT, and the penis remains in its flaccid/unstretched state throughout the night (Fig. 1a–c). Traditionally, penile tissue exhibits high rigidity and stretching during erection. In our study, during a penile erection, the intracavernous pressure (ICP) increased penis oncotic pressure and sustained erection, which indicated that the penis had high expansion pressure and rigidity during erection in control mice, while ED model mice exhibited a flaccid/unstretched penis and lost the mechanical stimulation (Fig. 1d, Fig. S1a). These clinical and experimental observations raise the question of whether sporadic stretching of the penis and its associated mechanical stimulation are necessary to maintain the function of penile erection.

To determine the pathways most prominently regulated by penile erection, we performed global transcriptome sequencing and KEGG pathway analyses on the erectile and flaccid penis of mice. We identified that cellular contraction, cytoskeleton organization, cAMP regulation, cGMP regulation, and the Hippo pathway were affected during penile erection (Fig. 1e, f). YAP and TAZ are downstream effectors of the Hippo pathway and regulate cell growth and plasticity during development and tissue regeneration[10–14]. YAP/TAZ is also an essential sensor through which cells detect the mechano-microenvironment of their surrounding tissue by mechanotransduction[15–18]. Here, we found that the expression levels of the YAP/TAZ transcriptional targets *Cyr61* and *Ptx3* in the erect penis were much higher than in the flaccid penis (Fig. 1g, h).

Next, to evaluate the changes in gene expression at single-cell resolution during the erection, we performed single nucleus RNA sequencing on flaccid ($n = 4744$ cells) and erect penis ($n = 5650$ cells) samples. Using graph-based clustering, we identified different cell populations, such as SWCs (schwann cells), SMCs (smooth muscle cells), PCs (pericytes), MACs (macrophages), FBs (fibroblasts), Epis (epithelial cells), and Endos (endothelial cells), based on canonical cell type marker gene expression (Fig. S1b, c). As expected, there were no significant changes in the cell components of the flaccid and erect penis (Fig. 1i). Next, we performed KEGG analysis to identify the most important changes in cellular function that occur during erection. Several pathways were altered during penile erection, including actin cytoskeleton-related pathways, ECM receptor-related pathways, the Hippo pathway, and cAMP and cGMP regulatory pathways (Fig. 1j). Among all identified cell types, SMCs caught our attention due to their central role in sustaining a penile erection. Then, through motif enrichment analysis based on differentially expressed gene sets between erection and flaccid conditions in SMCs, we identified the AP1 binding motif as a highly enriched regulatory element (Fig. S1d). It has been previously shown that AP1 and TEAD serve as YAP/TAZ transcriptional platforms, suggesting that YAP/TAZ may play an essential role during penile erection control[19]. Next, we analyzed the YAP/TAZ transcriptional target gene changes between flaccid and erection conditions from our single-nucleus RNA-seq data. GSEA and heatmap

results showed that the YAP/TAZ activity signature dramatically increased in penile SMCs during erection (Fig. 1k, l, S1e). Moreover, in situ hybridization (ISH) for *Cyr61* mRNA, as a canonical YAP/TAZ target gene, exhibited higher staining levels in the SMCs of the erect penis than in those of the flaccid penis, implying that YAP/TAZ in penile SMCs responds to mechanical stretch (Fig. 1m). Collectively, these data indicated that an erection stretches penile SMCs and activates the YAP/TAZ mechanoresponse.

### YAP/TAZ modulates erectile function and facilitates restoration in ED model mice

We have shown that an erection stretches SMCs, which is a mechano-signal that links the erection of the penis with YAP/TAZ activity. We next investigated the role of YAP/TAZ in normal erectile function. We asked whether maintaining YAP/TAZ activity in a healthy penis is essential for erectile function. To address this question, we used *Myh11Cre^ERT2^;Yap^fl/+^;Taz^fl/fl^* (*Taz* cKO), *Myh11Cre^ERT2^;Yap^fl/fl^;Taz^fl/+^* (*Yap* cKO) and *Myh11Cre^ERT2^;Yap^fl/fl^;Taz^fl/fl^* (*Y/T* cKO) mice to deplete YAP/TAZ from SMCs and identify the changes in erectile function. Interestingly, both *Taz* cKO and *Yap* cKO mice exhibited erectile dysfunction. Moreover, erectile function was abolished in *Y/T* cKO mice (Fig. 2a). Next, we used verteporfin to inhibit YAP/TAZ transcriptional activity in healthy male mice to identify changes in erectile function. The results showed that erectile function was impaired after 14 days of verteporfin treatment (Fig. 2b). We next assessed the YAP/TAZ activity states in patients suffering from ED. Compared to control samples, ED patient samples exhibited remarkably lower TAZ protein levels in the SMCs of the corpus cavernosum (Fig. 2c). These results suggest that the loss of YAP/TAZ activity could potentially lead to symptoms of erectile dysfunction (ED), highlighting the importance of YAP/TAZ in maintaining normal erectile function in the penis.

Next, we investigated the role of YAP/TAZ in penile rehabilitation following ED. To answer this question, we constructed an ED rat model. The two main clinical treatments for prostate cancer are radical prostatectomy and radiotherapy; however, approximately 50% of patients exhibit ED symptoms as a side effect[20,21]. We used the bilateral cavernous nerve crush injury (BCNI) rat model to mimic radical prostatectomy-induced ED[22]. First, we measured the ICP to identify the degree of erectile function after BCNI-induced ED. The ICP gradually decreased from 5 to 14 days and then recovered from 28 to 60 days after surgery (Fig. 2d upper panel and 2e). To understand the role of YAP/TAZ during BCNI-induced ED, we evaluated YAP/TAZ cellular localization and TAZ expression in penile SMCs of BCNI-treated rats at the indicated time points using immunohistochemistry (IHC) staining (Fig. 2d lower panel and f, Fig. S2a, b). Nuclear YAP/TAZ colocalization significantly decreased after BCNI from Day 5, reached the lowest point on Day 14, and was completely restored on Day 60. YAP/TAZ activity was highly associated with ED progression after BCNI treatment in rats. The YAP/TAZ transcriptional target genes *Ctgf* and *Cyr61* followed the same trend (Fig. 2g, Fig. S2c). Similar results were obtained using radiotherapy-induced ED[23] or castration-induced ED models[24] (Fig. 2h–m, Fig. S2d–f). Collectively, these data suggest that YAP/TAZ activity correlates with erectile function, and this correlation is also observed during recovery from experimentally induced ED. Next, we continuously treated mice with verteporfin three times per week to inhibit YAP/TAZ after the onset of ED by BCNI. We found that verteporfin-treated mice exhibited a strong and long-lasting ED phenotype instead of slow penile rehabilitation (Fig. 2n). These findings suggest that YAP/TAZ is involved in the regulation of normal erectile function and has the potential for promoting the recovery of erectile function during penile rehabilitation.

### Mechanical stretching exerts a predominant effect on YAP/TAZ activity in comparison to PDE5i treatment

During a penile erection, NO release increases cGMP concentrations and decreases intracellular $Ca^{2+}$ levels, leading to smooth muscle

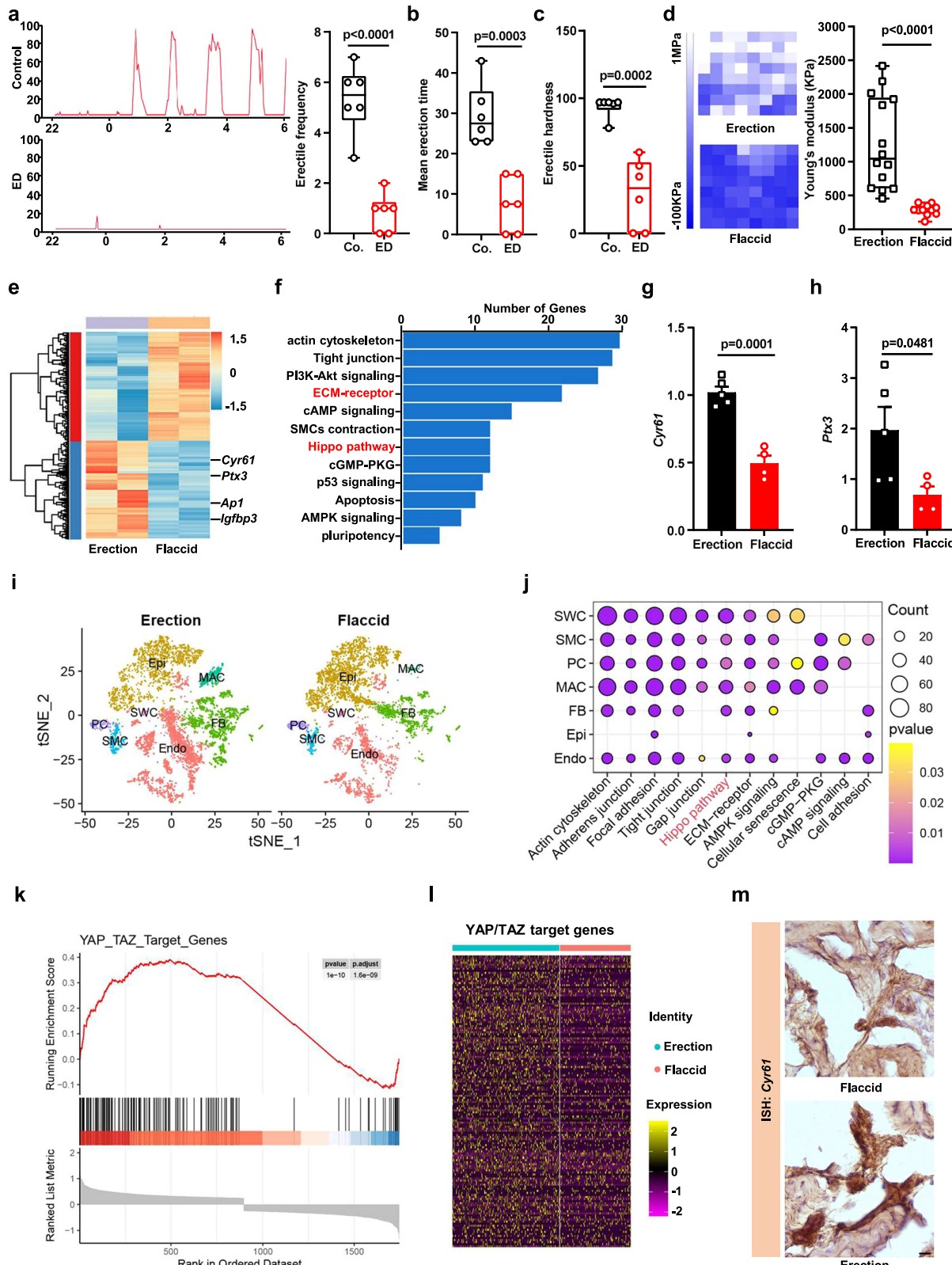

relaxation[1,2]. Subsequently, blood fills the corpus cavernosum to achieve an erection. In this physiological process, penile SMCs receive two mechanical signals. One is intracellular relaxation triggered by NO-cGMP signaling, and the other signal is mechanical stretch from the erect penis. Next, we wanted to determine the dominant signal regulating YAP/TAZ activity in penile SMCs. We first addressed whether YAP/TAZ can be regulated by mechanotransduction in penile SMCs.

We plated SMCs on stiff ECM (extracellular matrix) vs. soft ECM (0.25 Kpa) to mimic the high or low mechanics of SMCs and found that YAP/TAZ was excluded from the nucleus when SMCs were plated on soft ECM (Fig. 3a and Fig. S3a). Next, we treated SMCs with different mechanotransduction inhibitors, such as Src inhibitor (dasatinib), Rock1 inhibitor (Y27632), and F-actin inhibitor latrunculin A (Lat. A) to mimic a condition of low mechanical inputs. The results showed that

**Fig. 1 | An erect penis mechanically stretches SMCs and activates YAP/TAZ.**
Representative images of the NPT from healthy men (**a**, upper) and ED patients (**a**, bottom) during their sleep. The erectile frequency (**a**, right), mean erection time (**b**) and erectile hardness (**c**) between healthy men and ED patients were quantified. $n = 6$ biologically independent samples. **d**, AFM stiffness map images (left) and quantifications (right) of the stiffness from the flaccid penis ($n = 11$) and erectile penis ($n = 14$) of mice. Each point represents a detected area from 3 mice. **e**–**f** Heatmap and KEGG signaling pathways show the significantly changed genes and signaling pathways between the erectile penis and flaccid penis of mice. **g**, **h** The expression of the YAP/TAZ target genes *Cyr61* and *Ptx3* from the erectile penis ($n = 5$) and flaccid penis ($n = 4$) of mice. Data are presented as the mean ± sem.

**i** T-distributed stochastic neighbor embedding (t-SNE) plot of single cell under flaccid and erection conditions. **j** Representative KEGG signaling terms enriched in genes differentially expressed between flaccid and erection group for each cell type. **k** GSEA analyze of YAP/TAZ target genes between flaccid and erection conditions of SMCs. **l** Heatmap plots of YAP/TAZ target genes between flaccid and erection conditions in SMCs. **m** Representative ISH images of the YAP/TAZ target genes *Cyr61* from mice's flaccid and erectile penis, the experiments were repeated three independent times with similar results. Box plots indicate median (middle line), 25th, 75th percentile (box), minima, maxima and all points. The statistical analysis was calculated by two-sides unpaired Student's *t*-test, the confidence interval is 95%. Scale bars, 10 μm. Source data are provided as a Source Data file.

YAP/TAZ nuclear localization and transcriptional activity were significantly reduced when cells were not mechanically challenged (Fig. 3b–d, Fig. S3b). These results indicated that YAP/TAZ activity in penile SMCs was regulated by mechanotransduction.

To determine the relative influence of mechanical stretch and cytoskeletal contractility on YAP/TAZ activity, SMCs were cultured under conditions of high and low mechanical input, with and without the presence of PDE5i. We found that in sparsely plated SMCs (high mechanical input), YAP/TAZ activity was dominated by high mechanical signaling regardless of PDE5i presence. However, in densely plated SMCs (low mechanical input), PDE5i significantly reduced YAP/TAZ activity (Fig. S3c–f). To further examine the role of mechanical input, we also subjected densely plated SMCs to mechanical stretching, mimicking high cytoskeletal mechanics. The results of our experiment indicated that YAP/TAZ activity was primarily driven by the high mechanical input from SMC stretching, regardless of the presence or absence of PDE5i, as demonstrated in Fig. 3e–h. Moreover, we also modified the ECM stiffness to achieve the different cellular conditions of cell spreading and stretching in the presence or absence of PDE5i to induce SMC relaxation. We found that the localization of YAP/TAZ in the nucleus was mostly influenced by the mechanical stimulus from the ECM, rather than by cytoskeletal relaxation (as seen in Fig. 3i, j). Additionally, when SMCs displayed high mechanics, PDE5i was ineffective in altering YAP/TAZ activity. (Fig. 3k–l, Lat. A serves as a negative control by destroying the cellular F-actin). Next, we used the myosin inhibitor blebbistatin (Ble) to induce SMC relaxation. As expected, YAP/TAZ was excluded from the nucleus and lost transcriptional activity when explanted penile SMCs were treated with blebbistatin. However, by adding jasplakinolide (Jas), which induces the polymerization of actin monomers into F-actin to mimic stretched SMCs, to blebbistatin-treated cells, YAP/TAZ activity was rescued (Fig. 3m–p, Fig. S3g–j). Conversely, if SMCs were in an extremely unstretchable situation, such as treating SMCs with Lat. A, increasing cell contraction by adding a myosin activator (Myo A) was not sufficient to rescue YAP/TAZ activity (Fig. 3q–t, Fig. S3k–n).

Having shown above that YAP/TAZ loss-of-function triggered loss of erectile function, we next set out to test whether PDE5i could cure YAP/TAZ deficiency-induced ED. We treated *Yap* cKO and *Taz* cKO mice, which induced ED in mice but did not abolish erection as in *Y/T* cKO mice, with tadalafil (a PDE5i) to verify the effects of PDE5i on YAP/TAZ deficiency-induced ED. As a positive control, tadalafil increased erectile function in wild-type mice. However, tadalafil could not cure ED induced by loss of YAP or TAZ (Fig. 3u). Moreover, to further validate our findings, we employed a YAP/TAZ activator (PY60). Our results showed that PY60 effectively restored erectile function in the BCNI-induced ED model, a type of ED that does not respond to PDE5i treatment, as evidenced by the failure of tadalafil to produce a similar effect (Fig. 3v).

Collectively, these data indicated that erection-induced stretching of penile SMCs potently induced endogenous YAP/TAZ activity. Mechanical stretching dominated over the effect of intracellular contraction in determining YAP/TAZ activity in penile SMCs, as shown by the observation that PDE5i had no effect on YAP/TAZ deficiency-

induced ED and that the YAP/TAZ activator rescued BCNI-induced ED. Our findings may thus provide an explanation for why a substantial fraction of ED patients fail to respond to PDE5i therapy.

## ADM is transcriptionally regulated by mechano-YAP/TAZ and independent of PDE5i

YAP and TAZ are transcriptional coactivators that interact with the DNA-binding transcription factor TEAD (TEA domain family member) to recruit YAP/TAZ to induce their transcriptional activity[19,25–27]. To further investigate the YAP/TAZ downstream effectors that affect erection, we performed RNA-seq on YAP/TAZ-depleted SMCs. We found that adrenomedullin (ADM), a vasodilator peptide hormone synthesized by endothelial and smooth muscle cells[28,29], was downregulated by approximately 7-fold in YAP/TAZ knockdown cells, exhibiting a magnitude of downregulation similar to the well-known YAP/TAZ target genes *Cyr61*, *Ankrd1*, *Amotl2*, and *Ptx3*. This finding suggested that ADM is a potential YAP/TAZ transcriptional target (Fig. 4a, Fig. S4a, b). Further experiments confirmed that *Adm* expression levels decreased when SMCs were depleted of YAP/TAZ by either siRNA or genetic knockout (Fig. 4b, c, Fig. S4c). Importantly, ADM is also a potential target of YAP/TAZ according to the YAP1 Chip-seq database[30]. Mapping of YAP/TAZ binding to the ADM promoter (Fig. S4d, e) revealed that YAP binds to a region within 557-699 base pairs upstream of the TSS of ADM (Fig. 4d). These results suggested that ADM is a direct transcriptional target of YAP/TAZ in SMCs. We thus asked whether YAP/TAZ activity would correlate with ADM levels in vivo. Indeed, we found that *Adm* expression levels significantly decreased in penile SMCs from *Y/T* cKO mice compared to control animals (Fig. 4e). Next, we measured the expression of ADM in our established ED models to investigate whether ADM would be consistently downregulated in ED-affected tissues. Indeed, in both BCNI-induced and irradiation-induced ED models, *Adm* expression levels correlated with YAP/TAZ activity and erectile function (Fig. 4f–i).

We further investigated whether ADM expression levels are controlled by cellular mechanics and correlated with YAP/TAZ activity. We found markedly downregulated *Adm* expression when SMCs exhibited low mechanics or were treated with dasatinib, cerivastatin, Y27632, blebbistatin, and Lat. A. These results suggested that mechanotransduction controls *Adm* expression (Fig. 4j–l).

Next, we stretched the SMCs or modified the ECM stiffness to simulate different cellular stretching conditions. Concomitantly, we treated SMCs with either vehicle or PDE5i to induce SMC relaxation. We found that extracellular stretching potently induced YAP/TAZ binding to the *Adm* promoter and was the main regulator of *Adm* expression, overruling PDE5i-mediated cytoskeletal relaxation (Fig. 4m–p). Next, we aimed to dissect the influence of F-actin network polymerization versus cell contractility on the YAP/TAZ-Adm mechanoresponse in SMCs. When SMCs were treated with blebbistatin (Ble), *Adm* was downregulated. However, concomitant treatment with jasplakinolide (Jas) to mimic a stretched cell state rescued *Adm* expression. Conversely, if SMCs were treated with Lat. A, increasing SMC contraction by adding a myosin activator (Myo A) could not rescue *Adm* expression (Fig. 4q, r).

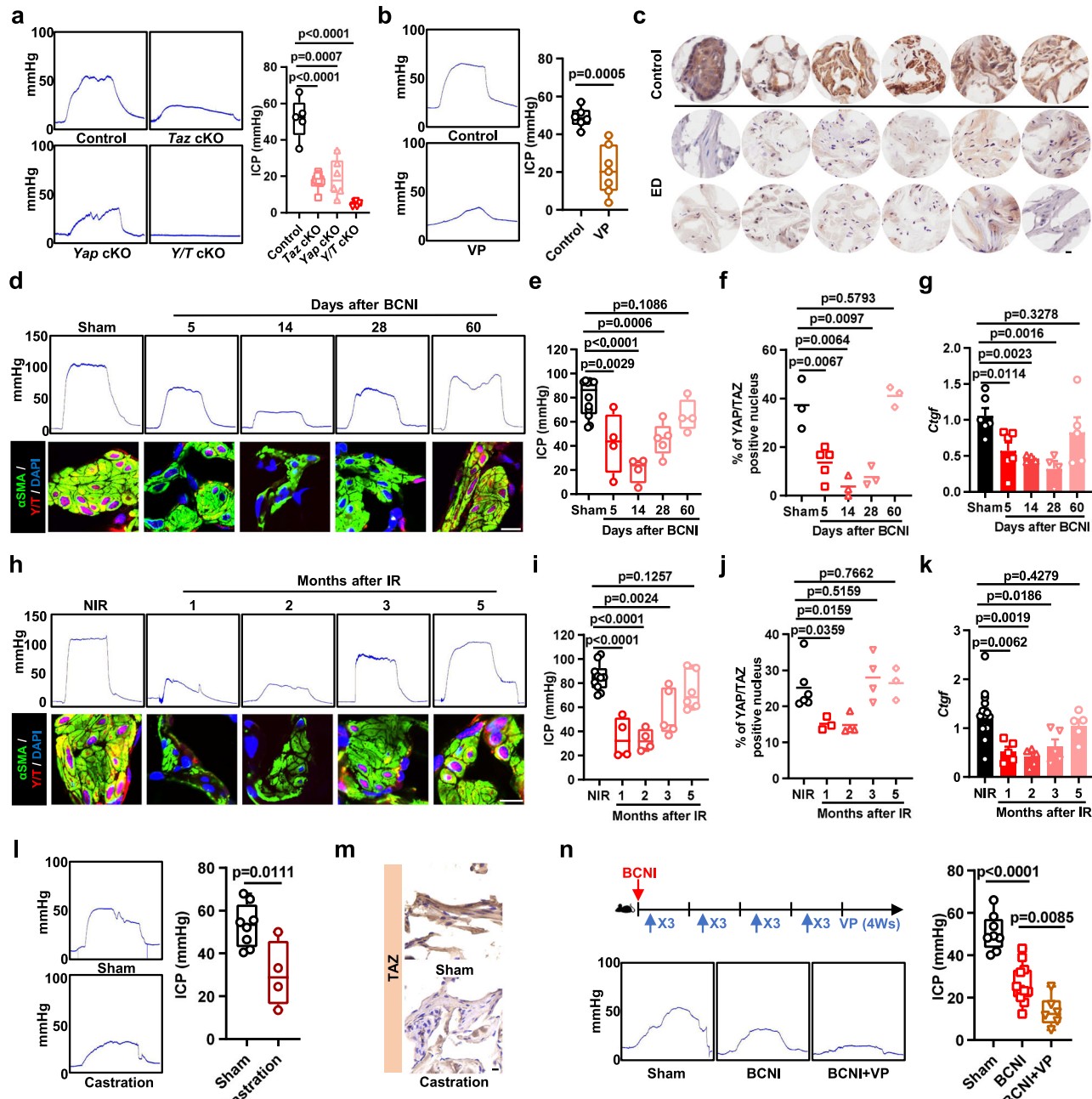

**Fig. 2 | YAP/TAZ modulates erectile function and facilitates restoration in ED model mice. a** ICP from wild-type (Control, $n=5$), *Myh11Cre^ERT2^; Yap^fl/+^; Taz^fl/fl^* (*Taz* cKO, $n=10$), *Myh11Cre^ERT2^; Yap^fl/fl^; Taz^fl/+^* (*Yap* cKO, $n=6$) and *Myh11Cre^ERT2^; Yap^fl/fl^; Taz^fl/fl^* (Y/T cKO, $n=5$) mice. **b** ICP from the control ($n=6$) and verteporfin-treated ($n=7$) mice. **c** Immunohistochemistry images of TAZ from control and ED patients, each image represents a human sample, the experiments were repeated three independent times with similar results. **d** ICP (upper panel) and immuno-fluorescence images (bottom panel) of YAP/TAZ from sham rats and the indicated time points of the BCNI-induced rat ED models. **e** Quantification of the ICP in sham rats ($n=12$) and 5 days ($n=4$), 14 days ($n=4$), 28 days ($n=5$) and 60 days ($n=4$) indicated time points of the BCNI-induced rat ED models. **f** Quantification of immunofluorescence images of YAP/TAZ (red), αSMA (green) and DAPI (blue) from sham rats ($n=3$) and 5 days ($n=5$), 14 days ($n=3$), 28 days ($n=3$) and 60 days ($n=3$) indicated time points of the BCNI-induced rat ED models. **g** Expression of the YAP/TAZ target gene *Ctgf* from sham rats ($n=6$) and 5 days ($n=6$), 14 days ($n=4$), 28 days ($n=4$) and 60 days ($n=5$) indicated time points of the BCNI-induced rat ED models. Similar results were obtained by checking *Cyr61* expression (Supplementary Fig. 2c). **h** ICP (upper panel) and immunofluorescence images (bottom panel) of YAP/TAZ from the control and indicated time points of the IR-induced rat ED models. **i** Quantification of the ICP in control ($n=10$) and 1 month ($n=4$), 2 months

($n=4$), 3 months ($n=5$) and 5 months ($n=6$) indicated time points of the IR-induced rat ED models. **j** Quantification of immunofluorescence images of YAP/TAZ (red), αSMA (green) and DAPI (blue) from control ($n=6$) and 1 month ($n=3$), 2 months ($n=4$), 3 months ($n=4$) and 5 months ($n=3$) indicated time points of the IR-induced rat ED models. **k** Expression of the YAP/TAZ target gene *Ctgf* from control ($n=15$) and 1 month ($n=5$), 2 months ($n=5$), 3 months ($n=5$) and 5 months ($n=5$) indicated time points of the IR-induced rat ED models. Similar results were obtained by checking *Cyr61* expression (Supplementary Fig. 2h). **l** ICP and quantifications from sham ($n=8$) and castration ($n=4$) mice. **m** Immunohistochemistry images of TAZ from sham and castration mice, the experiments were repeated three independent times with similar results. **n** Upper panel, schematics of the corresponding experimental design. Representative images (bottom) and quantifications (right) of the ICP from the sham- ($n=8$), BCNI- ($n=11$), and verteporfin-treated ($n=6$) BCNI mice. Box plots indicate median (middle line), 25th, 75th percentile (box), minima, maxima and all points. Dot plots indicate mean (middle line) and all points. Bar charts are presented as the mean ± sem. The statistical analysis was calculated by two-sides unpaired Student's *t* test, the confidence interval is 95%. Each point represents a mouse or rat biologically independent sample. Scale bars, 10 μm. Source data are provided as a Source Data file.

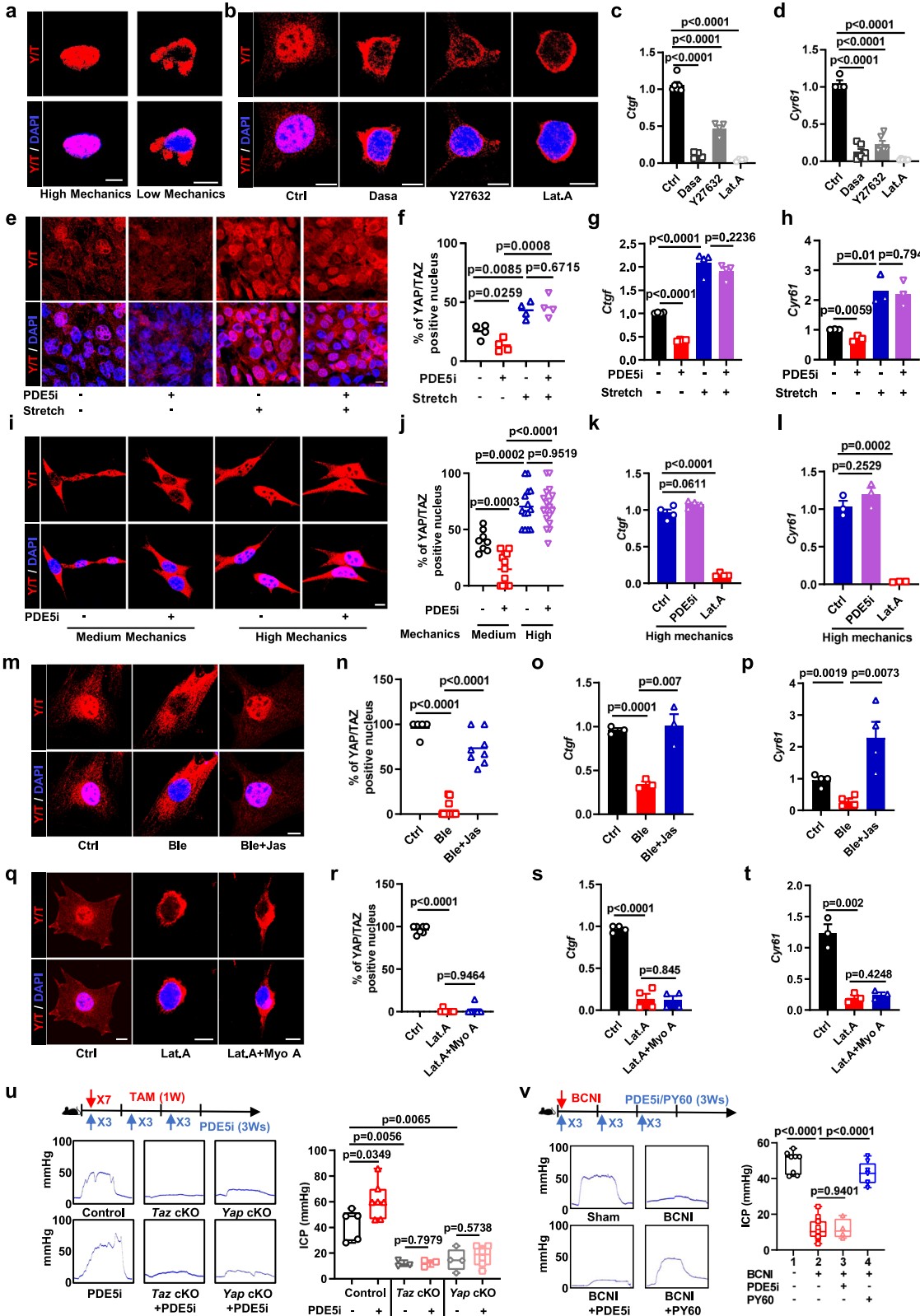

These experiments suggest that as a direct YAP/TAZ-regulated gene, *Adm* is under mechanotransduction control, mechanical stretch and F-actin network polymerization dominate over intracellular contraction in controlling *Adm* expression, and this regulation is independent of PDE5i.

## YAP/TAZ-ADM controls penile SMC contraction

Next, we aimed to identify how the mechano-YAP/TAZ-ADM pathway sustains penile erection function. Several previous studies have suggested that ADM directly increases intracellular cAMP levels in smooth muscle cells and stimulates NO release by binding to the Clr receptor

**Fig. 3 | Mechanical stretching exerts a predominant effect on YAP/TAZ activity in comparison to PDE5i treatment. a** Immunofluorescence of YAP/TAZ (red) and DAPI (blue) in SMCs replated on a high or low mechanics ECM. **b** Immunofluorescence of YAP/TAZ (red) and DAPI (blue) in SMCs from control or treated with the indicated mechanotransduction inhibitors, such as Dasa (dasatinib), Y27632, and latrunculin A (Lat.A). **c** Expression of the YAP/TAZ target gene *Ctgf* in SMCs from control ($n = 6$) or treated with the indicated mechanotransduction inhibitors, such as Dasa (dasatinib, $n = 4$), Y27632 ($n = 4$), and latrunculin A (Lat.A, $n = 4$). **d** Expression of the YAP/TAZ target gene *Cyr61* in SMCs from control ($n = 4$) or treated with the indicated mechanotransduction inhibitors, such as Dasa (dasatinib, $n = 6$), Y27632 ($n = 6$), and latrunculin A (Lat.A, $n = 6$). Immuno-fluorescence (**e**) of YAP/TAZ (red) and DAPI (blue), quantification (**f**, $n = 4$) of YAP/TAZ localization and expression of the YAP/TAZ target gene *Ctgf* (**g**, $n = 4$) and *Cyr61* (**h**, $n = 3$) in dense states SMCs treated with PDE5i in stretch/nonstretch treatment. Immunofluorescence (**i**) of YAP/TAZ (red) and DAPI (blue) and quanti-fication (**j**) of YAP/TAZ localization in SMCs treated with PDE5i in medium (PDE5i untreated, $n = 8$; PDE5i treated, $n = 13$) or high (PDE5i untreated, $n = 14$; PDE5i treated, $n = 18$) mechanical ECM. Expression of the YAP/TAZ target gene *Ctgf* (**k**, $n = 4$) and *Cyr61* (**l**, $n = 3$) in SMCs treated with the latrunculin A (Lat.A) on the

basis of high mechnical ECM. Immunofluorescence (**m**) of YAP/TAZ (red) and DAPI (blue), quantification (**n**) of YAP/TAZ localization in SMCs from control ($n = 5$) and treated with Ble ($n = 13$) or combination with Jas ($n = 8$). Expression of the YAP/TAZ target gene *Ctgf* (**o**, $n = 3$) and *Cyr61* (**p**, $n = 4$) in SMCs treated with Ble or Jas. Immunofluorescence (**q**) of YAP/TAZ (red) and DAPI (blue), quantification (**r**) of YAP/TAZ localization in SMCs from control ($n = 7$) treated with Lat.A ($n = 7$) or combination with Myo A ($n = 16$). Expression of the YAP/TAZ target gene *Ctgf* (**s**, $n = 4$) and *Cyr61* (**t**, $n = 3$) in SMCs treated with Lat.A or Myo A. **u** ICP from the control and PDE5i (Tadalafil) treated (PDE5i untreated, $n = 5$; PDE5i treated, $n = 7$), Taz cKO (PDE5i untreated, $n = 3$; PDE5i treated, $n = 3$) and Yap cKO (PDE5i untreated, $n = 4$; PDE5i treated, $n = 6$) mice. **v** ICP from the Sham ($n = 8$), BCNI ($n = 15$), BCNI + PDE5i (Tadalafil, $n = 4$), BCNI + PY60 (YAP/TAZ activator, $n = 8$) mice. Box plots indicate median (middle line), 25th, 75th percentile (box), minima, maxima and all points. Dot plots indicate mean (middle line) and all points. Bar charts are presented as the mean ± sem. The statistical analysis was calculated by two-sides unpaired Student's $t$ test, the confidence interval is 95%. The point represents a mouse or detected cell area over 3 biologically independent experiments with similar results. Scale bars, 10 μm. Source data are provided as a Source Data file.

(calcitonin-like receptor) and Ramp2/3 receptor (receptor activity modifying protein 2/3), leading to smooth muscle relaxation[28,29,31,32]. To investigate how the YAP/TAZ-ADM axis regulates erectile function, we explored the effects of YAP/TAZ-ADM on SMC contraction. We found that YAP/TAZ KO SMCs exhibited higher contractile potential when embedded into collagen gels (Fig. 5a). We also found that cellular $Ca^{2+}$ concentrations and pMLC levels significantly increased upon YAP/TAZ or ADM depletion, which suggested that YAP/TAZ-ADM feeds back onto actomyosin contractility in penile SMCs (Fig. 5b, c, Fig. S5a–d). Moreover, knockdown of either YAP/TAZ or ADM increased collagen gel contraction, demonstrating that YAP/TAZ-ADM loss increases basal SMC contractility (Fig. S5e).

We next investigated the epistatic relationship between YAP/TAZ and ADM in regulating the erectile response. If ADM acts downstream of YAP/TAZ in regulating SMC relaxation, its sustained expression should rescue the effect of YAP/TAZ loss on SMC contractility. In line with this epistatic relationship, we found that ADM overexpression in YAP/TAZ knockout SMCs could restore the cellular $Ca^{2+}$ concentrations, pMLC levels and contractile ability (Fig. 5d–g, Fig. S5f–i). Conversely, in keeping with ADM expression operating downstream of YAP/TAZ in regulating SMC contraction, experimentally sustaining YAP/TAZ in an ADM loss-of-function context should be incon-sequential. Indeed, we found that ectopic expression of TAZ in ADM knockdown cells did not change the contractile capacity of SMCs compared to that of ADM-deficient cells (Fig. S5j). Together, these results indicated that YAP/TAZ regulated ADM to affect SMC con-traction ability.

Finally, we verified our observations in vivo. We asked whether reconstituting ADM levels in vivo could rescue ED induced by YAP/TAZ depletion. Yap deficiency-induced ED model mice were established by injecting *Myh11Cre^ERT2;Yap^fl/fl;Taz^fl/+* (*Yap* cKO) mice with tamoxifen. Mice were split into two groups and treated with either vehicle or ADM (2.4 μmol/kg) 3 times per week for two weeks via tail vein injection. We also treated wild-type mice with either vehicle or ADM as a control and reference point. We then measured the erectile function of the mice by ICP 14 days after depleting YAP in the mice. ADM injection significantly increased erectile function in wild-type mice, suggesting that ADM affects erectile function. Moreover, treatment with ADM partially res-cued the erectile function of *Yap* cKO mice, which were previously found to be resistant to standard PDE5i therapy (Fig. 5h, i and Fig. 3u). Similarly, treatment with ADM also rescued mouse erectile function in the BCNI-induced ED model, another type of PDE5i-resistant ED (Fig. S5k–m). These results collectively suggest that restoring ADM levels can normalize SMC relaxation and improve YAP/TAZ deficiency- or BCNI-induced ED in vivo.

Penile erection is mediated by smooth muscle cell (SMC) relaxa-tion induced by NO-cGMP activation. Our work identified a model that explains how this process is sustained. Penile erection causes stretching of penile SMCs with subsequent YAP/TAZ activation. Nuclear YAP/TAZ directly regulates the transcription of adrenome-dullin (ADM), a local actin hormone controlling vascular tone. ADM diffuion leads to the smooth muscle relaxation required to support and maintain an erection (Fig. 5j).

## Mechanotransduction promotes PDE5i-nonresponding ED recovery by increasing YAP/TAZ-ADM activity

Our study has demonstrated the involvement of the mechano-YAP/TAZ-ADM axis in SMCs in the maintenance of erectile function and its potential role in ED recovery. Next, we used clinically established physiotherapies, vacuum erection devices (VEDs), and shock wave therapy (SWT) to detect whether the improvements in erectile func-tion achieved by these intervention measures also rescued YAP/TAZ activity. We used VED therapy to stimulate erectile function 3 times per day to mimic NPT in the healthy penis of rats in our BCNI-induced ED model, as a disease model in which PDE5i has a negligible effect, recapitulating PDE5i irresponsiveness observed in clinical settings within a subgroup of ED patients (Fig. S6a). The ICP of the VED-treated mice was significantly increased compared to that of nontreated BCNI-ED rats, indicating that VED can promote the recovery of BCNI-induced ED (Fig. 6a, b). This increased ICP was correlated with an increase in YAP/TAZ activity. Indeed, the YAP/TAZ target genes *Ctgf* and *Cyr61* were considerably upregulated in the VED treatment group compared to the nontreated BCNI-ED group (Fig. 6c, Fig. S6b). We also found that TAZ protein was downregulated after the onset of ED but selectively restored after VED treatment (Fig. 6d). IHC and IF for YAP/TAZ yielded consistent results (Fig. 6e, Fig. S6c–e). Moreover, in vivo, VED also promoted *Adm* expression in the ED model rats (Fig. 6f, g). As an independent corroboration, we employed SWT as another example of a widely used clinical therapy for ED to treat BCNI-induced ED in rats. The mechanism of SWT has been suggested to be based on the mechanical stimulation of cells, thereby activating mechan-otransduction pathways[33–35]. Consistently, we found that SWT can reverse ED by increasing YAP/TAZ-ADM activity (Fig. 6h–n, Fig. S7a–d). To eliminate the influence of SMC-specific YAP/TAZ deletion on the vasculogenic system, we locally injected AAV-Cre and adenovirus-Cre into the penis of *Yap^fl/fl;Taz^fl/fl* mice to create penis-specific knockout mice (Fig. S7h). Compared to the control mice, both of the penis-specific knockout mice exhibited erectile dysfunction (ED), as demonstrated in Fig. 6o–p, Columns 5 and 7. Using these animal models, we tested the hypothesis that the YAP/TAZ-ADM pathway

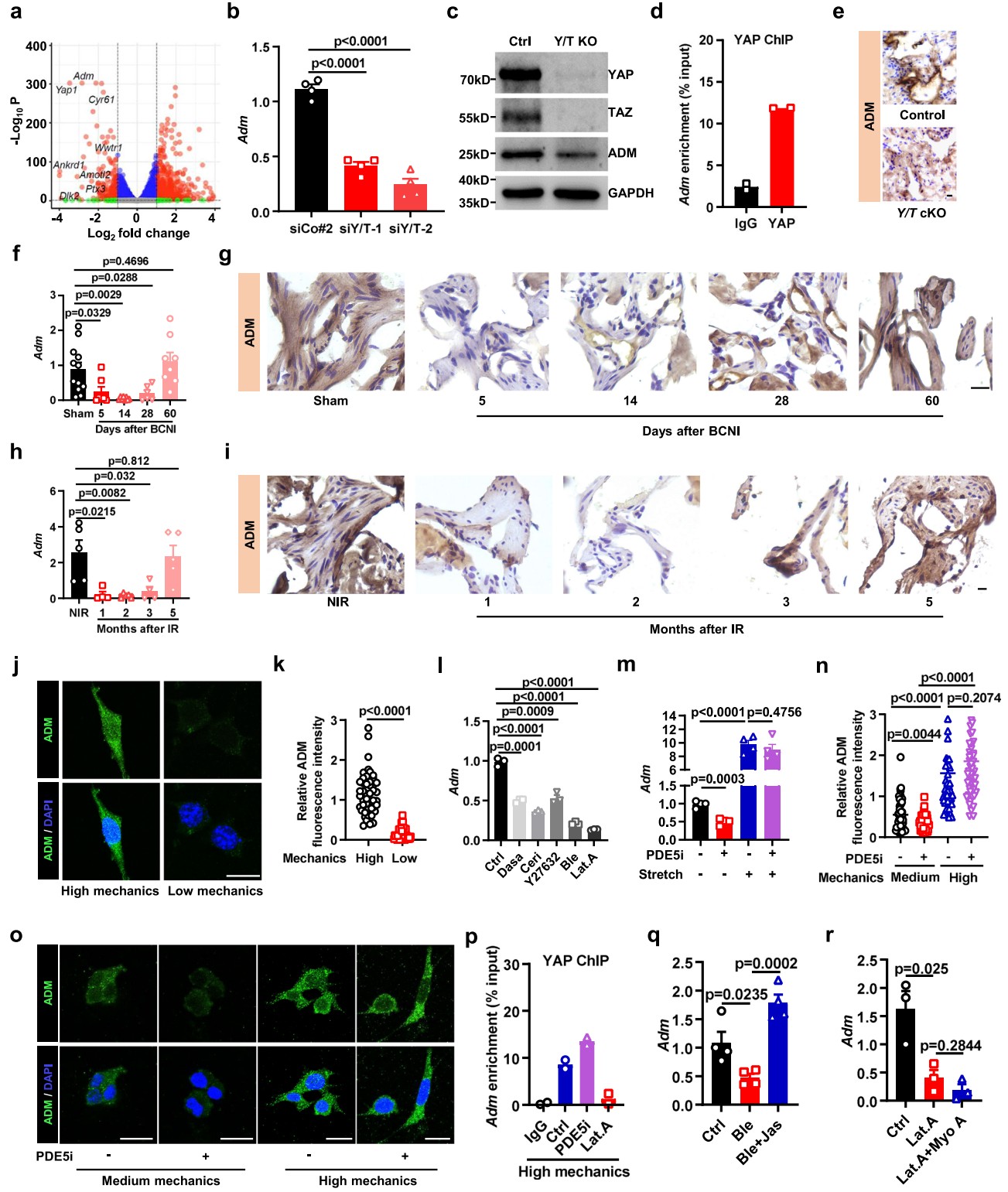

mediates the impact of mechanical stimuli on penile erection. As a result, supplementary mechanical stimuli are unlikely to provide a cure for ED in YAP/TAZ-deficient mice. As shown in Fig. 6o–p, we demonstrated that VED failed to produce a curative effect in either SMC-specific or penile-specific YAP/TAZ-deficient ED model mice.

We conducted this study in response to clinical observations indicating that ED patients lose their regular erection during sleep, while healthy males experience nocturnal penile tumescence (Fig. 1a). We thus aimed to test whether our observations that mechanical

stimulation improves ED, especially in PDE5i nonresponders, also holds true in clinical settings in human patients. We tested whether stimulating erection VED might have a curative effect on ED patients, especially in PDE5i nonresponders. We collected and analyzed 73 ED patients from 18 to 80 years of age, dividing them into four groups based on their medical experience: patients who are PDE5i non-responders (19.18%); patients who experience side effects to PDE5i and refuse to take PDE5i (43.84%); patients who are afraid of PDE5i addiction and refuse to take PDE5i (19.18%); and patients who have no prior

**Fig. 4 | ADM is transcriptionally regulated by mechano-YAP/TAZ axis and independent of PDE5i.** a Volcano plot showing the altered genes in primary penile SMCs treated with siYAP/TAZ vs siCo. b The expression of *Adm* in primary penile SMCs treated with siYAP/TAZ. Data are the mean ± sem of *n* = 4 biologically independent samples. c Western blots were used to assess the expression of YAP, TAZ and ADM in primary penile SMCs derived from *Yap^{fl/fl}::Taz^{fl/fl}* mice treated with Ad-Cre to induce YAP/TAZ knockout, the experiments were repeated three independent times with similar results. d ChIP-PCR assesses the interaction between YAP and the ADM promotor in SMCs, *n* = 2 biologically independent samples. e Immunohistochemistry assessing the expression of *Adm* from wild-type and *Y/T* cKO mice, the experiments were repeated three independent times with similar results. f, g qRT-PCR and Immunohistochemistry assessed the expression of *Adm* from sham rats (*n* = 12) and 5 days (*n* = 7), 14 days (*n* = 8), 28 days (*n* = 6) and 60 days (*n* = 8) indicated time points of the BCNI-induced rat ED models. h, i qRT-PCR and Immunohistochemistry assessed the expression of *Adm* from control (*n* = 5) and 1 month (*n* = 4), 2 months (*n* = 5), 3 months (*n* = 4) and 5 months (*n* = 5) indicated time points of the IR-induced rat ED models. j, k Immunofluorescence of ADM (green) and DAPI (blue) in SMCs replated on a high stretch (*n* = 48) or low stretch

(*n* = 42) ECM, each point represents an area examined over 3 independent experiments. l qRT-PCR assessing the expression of *Adm* in SMCs treated with the indicated mechanotransduction inhibitors (*n* = 3). m qRT-PCR assessed the expression of *Adm* in SMCs treated with PDE5i or stretch when plated in dense states (*n* = 4). Immunofluorescence (o) and quantification (n) of ADM (green) in SMCs treated with PDE5i in medium (PDE5i untreated, *n* = 45; PDE5i treated, *n* = 46) or high (PDE5i untreated, *n* = 51; PDE5i treated, *n* = 53) stretch ECM, each point represents an area examined over 3 independent experiments. p ChIP-PCR assesses the interaction between YAP and the ADM promotor in SMCs treated with the latrunculin A (Lat.A) or PDE5i on the basis of high stretch ECM (*n* = 2 biologically independent samples). q Expression of *Adm* in SMCs treated with Ble or Jas (*n* = 4). r Expression of *Adm* in SMCs treated with Lat.A and Myo A (*n* = 3). Dot plots indicate mean (middle line) and all points. Bar charts are presented as the mean ± sem. The statistical analysis was calculated by two-sides unpaired Student's *t* test, the confidence interval is 95%. Each experiment was repeated three independent times with similar results. The point represents a rat or detected cell area over 3 biologically independent experiments with similar results. Scale bars, 20 μm. Source data are provided as a Source Data file.

history with PDE5i (17.81%). To assess the curative effect, we asked patients to complete the five-item International Index of Erectile Function (IIEF-5) questionnaire before and after VED treatment. The clinical results demonstrated significant associations between the effects of VED and improvements in penile erectile function, regardless of PDE5i responsiveness. These findings indicate correlations between mechanostimulation and erectile function (Fig. 6q).

In conclusion, using two clinically established therapy options to induce penile mechano-stretching, we found that mechano-stimulatory therapeutic intervention improved penile rehabilitation. This improvement correlated with YAP/TAZ activity in penile SMCs. Finally, we tested our hypothesis using a clinical observational study involving VED treatment and identified associations between mechanostimulation and erectile function. Our findings indicate a positive correlation, showing that therapeutic intervention through mechano-stimulation associates with improved erectile dysfunction in patients, regardless of PDE5i usage.Thus, these lines of evidence strongly suggest that in penile erectile function, mechano-YAP/TAZ-ADM signaling in penile SMCs is epistatic to NO-cGMP signaling.

## Discussion

Erectile dysfunction (ED) is a common disease that typically affects middle-aged or elderly men and impacts their well-being and that of their partners[4]. Historically, the regulation of penile erection has been attributable to NO-cGMP signaling, which controls the contraction and relaxation of SMCs[2]. PDE5 inhibitors are commonly used as a first-line treatment for ED, but nonresponders require alternative options[3,5]. This study identified another regulator of erections, that is, blood inflow filling the corpus cavernosum after penile SMC relaxation caused by NO-cGMP activation provides a mechanical stimulus by stretching penile SMCs, which activates the YAP/TAZ-ADM axis to sustain penile SMC relaxation. In this model, the mechano-YAP/TAZ pathway serves as a regulator of penile erection. Indeed, we have shown that PDE5is are insufficient to rescue ED caused by genetic YAP/TAZ deficiency (Fig. 3u). In line with this collaborative relationship, our experimental mice and rat models, as well as the subsequent clinical observational study, revealed that mechano-stimulatory therapeutic interventions effectively alleviated erectile dysfunction (ED). It is important to note that these interventions demonstrated the ability to improve erectile function in animal models that were nonresponsive to PDE5i treatment. Additionally, they were associated with promoting erectile function in patients, irrespective of PDE5i treatment, as observed in the clinical observational study. (Fig. 6). Hence, these lines of evidence suggest that the intervention approaches could be aimed at either promoting mechanostimulation or YAP/TAZ activation itself to support erectile function.

In contrast to the plethora of promising small compound YAP/TAZ inhibitors in cancer therapy research, few YAP/TAZ activators have been identified to date. This lack of interest might be in part due to the potent oncogenic properties of YAP/TAZ[36–38], which might also limit the application of YAP/TAZ activators in ED therapy. However, given the broad interest in investigating YAP/TAZ inhibitors for cancer treatment, our research suggests that excessive inhibition of YAP/TAZ may lead to ED, which should be taken into consideration. Here, we found that ADM, a downstream target of YAP/TAZ, promotes erectile function, and may be another, potentially less risky, ED therapy target. Another potential benefit concerns the pharmacokinetics of ADM itself. As a peptide hormone (usually secreted by endothelial and smooth muscle cells), artificially resupplied ADM could diffuse freely between the bloodstream and interstitium. Notably, we observed that ADM could reverse ED caused by YAP/TAZ deficiency (Fig. 5h), making ADM a convenient prospective drug to treat PDE5i-nonresponsive ED patients.

A limitation of our study is that it primarily focuses on neurogenic ED that is nonresponsive to PDE5i therapy and does not examine the complexities of vascular ED, such as diabetic ED. Recent reports suggest that diabetic ED may increase YAP levels, which seems to contrast our findings[39,40]. We postulate that this discrepancy may be due to differences in sample isolation methods, such as single-cell RNA-seq from isolated live cells, which has the potential to overlook important mechanical information during sample preparation. Additionally, although animal models are commonly used to gain insights into disease mechanisms, there are still some species differences that may influence our results. Moreover, it is possible that the underlying mechanisms of ED in diabetic patients and those without diabetes may differ due to distinct pathogenic mechanisms. These observations suggest that further studies are needed to better understand the complex molecular and cellular mechanisms underlying different forms of ED and how these mechanisms may vary across patient populations. In the present study, we performed single-nucleus RNA sequencing analysis on gene expression profiles in both endothelial cells and pericytes during an erection. Our results indicate changes in the Hippo pathway and YAP/TAZ activity in both cell types (Fig. S8a, b), but the absence of significant changes in YAP/TAZ activity in endothelial cells led us to concentrate on pericytes. Our findings suggest a close link between YAP/TAZ activity in pericytes and ED progression. Furthermore, compared to untreated ED rats, rats treated with mechanical stimuli showed a significant increase in pericyte YAP/TAZ activity (Fig. S8c–h), indicating that the mechano-YAP/TAZ cascade is also active in pericytes. These findings lay the foundation for future research on the potential of the mechano-YAP/TAZ-adrenomedullin pathway as a target for ED treatment.

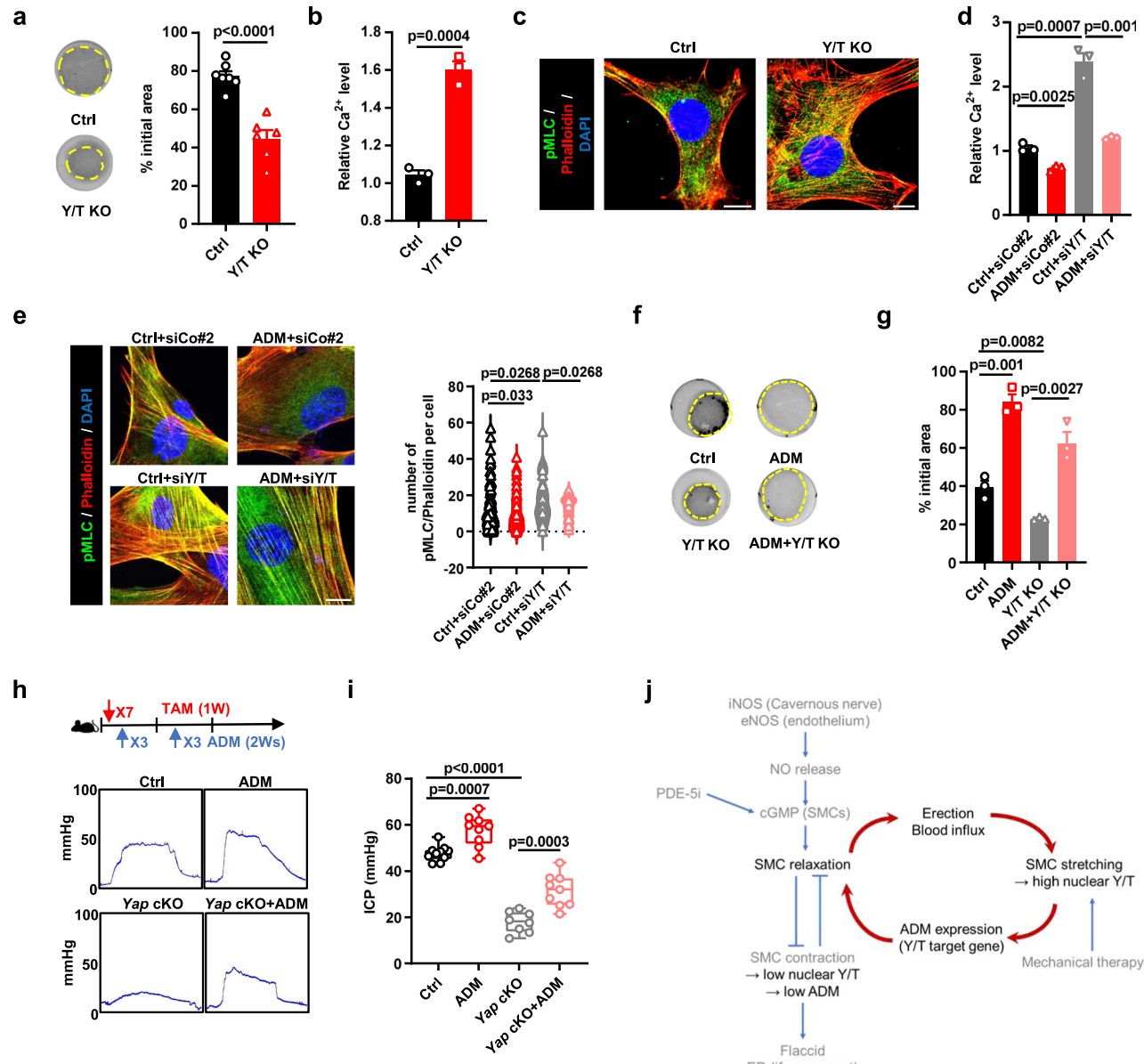

**Fig. 5 | YAP/TAZ-ADM controls penile SMC contraction. a** Representative contraction images (left) and quantifications of contraction ability (right) in primary penile SMCs treated with Ad-Cre to knock out YAP/TAZ (Ctrl, $n = 7$; $Y/T$ KO, $n = 6$). **b** Quantifications of Ca²⁺ staining in primary penile SMCs treated with Ad-Cre to knock out YAP/TAZ ($n = 3$). **c** Representative image of pMLC (red), phalloidin (green) and DAPI (blue) in primary penile SMCs treated with Ad-Cre to knock out YAP/TAZ. **d** Quantifications of Ca²⁺ staining in primary penile SMCs overexpressing Flag-ADM or knocking down YAP/TAZ ($n = 3$). **e** Representative immunofluorescence images (left) and quantifications (right) of pMLC (green), phalloidin (red) and DAPI (blue) in control ($n = 94$) primary penile SMCs, overexpressing Flag-ADM ($n = 83$), knocking down YAP/TAZ ($n = 47$) or combination group ($n = 53$), each point represents an area examined over 3 independent experiments.

**f, g** Contraction ability in primary penile SMCs overexpressing Flag-ADM or knocking out YAP/TAZ ($n = 3$). **h, i** Schematics of the corresponding experimental design. ICP from the control (PBS treated, $n = 10$; ADM treated, $n = 9$), and $Yap$ cKO (PBS treated, $n = 7$; ADM treated, $n = 9$) mice treated with PBS or ADM. **j** Model summarizing the main findings of this study. Box plots indicate median (middle line), 25th, 75th percentile (box), minima, maxima and all points. Violin and dot plots indicate mean (middle line) and all points. Bar charts are presented as the mean ± sem. The statistical analysis was calculated by two-sides unpaired Student's $t$-test, the confidence interval is 95%. The point represents a mouse or detected cell area over 3 biologically independent experiments with similar results. Scale bars, 10 μm. Source data are provided as a Source Data file.

## Methods
### Clinical NPT analysis and Clinical observational study for VED therapy of ED patients

Clinical retrospective observational data were obtained with written informed consent from male individuals with an age range between 18 and 80 years from Department of Urology, Affiliated Hospital of Changchun University of traditional Chinese Medicine during 2020.1-2021.12. The study was approved by the Affiliated Hospital of Changchun University of traditional Chinese Medicine ethics committee and complied with all relevant ethical regulations (License No. CCZY-FYLL2019sz.063). The severity of erectile dysfunction is often described as mild, moderate or severe according to the five-item International Index of Erectile Function (IIEF-5) questionnaire, with a score of 1–7 indicating severe, 8–11 moderate, 12–16 mild-moderate, 17–21 mild and 22–25 no erectile dysfunction.

The patients' medical experience and information, including their diagnosed duration in ED, whether they had been treated with PDE5i, and how the treatment was affected, were recorded. Meanwhile, all the

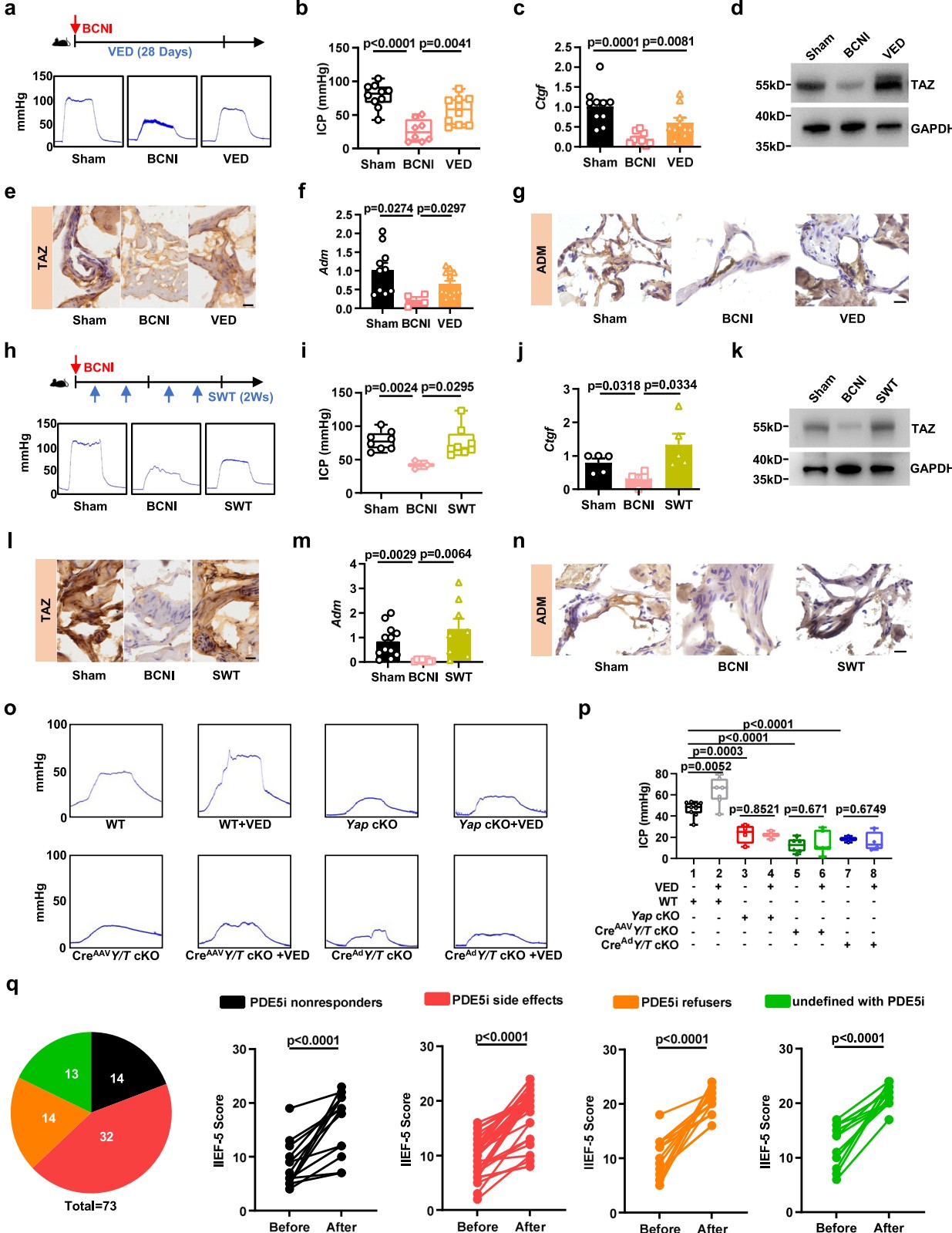

patients were asked to evaluate their erectile function on the IIEF-5 questionnaire before the treatments, which is the internationally recognized erectile function scoring standard. After the standard treatment for one month by the FDA-proved VED method, these patients were asked to re-evaluate their erectile function through the IIEF-5 questionnaire again.

Next, we divided them into four groups according to their medical experience: patients who are PDE5i non-responders; patients who experience side effects of PDE5i and refuse to take PDE5i; patients who are afraid of PDE5i addiction and refuse to take PDE5i; and patients who have no prior history with PDE5i. Last, we analyzed the erectile function of these patients by IIEF-5 scores before and after VED treatments.

**Fig. 6 | Mechanotransduction promotes PDE5i-nonresponding ED recovery by increasing YAP/TAZ-ADM activity.** Schematics of the corresponding experimental design (upper panel of **a**). Representative images (bottom panel of **a**) and quantifications (**b**) of the ICP from the sham ($n = 10$), BCNI ($n = 8$), and BCNI combined VED (VED)-treated rats ($n = 10$). The expression of the YAP/TAZ target gene *Ctgf* (**c**) from the sham ($n = 10$), BCNI ($n = 8$), and BCNI combined VED (VED)-treated rats ($n = 10$), western blots assessing the expression of TAZ (**d**) and Immunohistochemistry of TAZ (**e**). qRT-PCR (**f**) and Immunohistochemistry (**g**) assessing the expression of *Adm* from sham ($n = 10$), BCNI ($n = 4$), and BCNI combined VED (VED)-treated rats ($n = 12$). Schematics of the corresponding experimental design (upper panel of **h**). Representative images (bottom panel of **h**) and quantifications (**i**) of the ICP from the sham ($n = 8$), BCNI ($n = 3$), and BCNI combined SWT (SWT)-treated rats ($n = 8$). The expression of the YAP/TAZ target gene *Ctgf* (**j**) from the sham ($n = 5$), BCNI ($n = 4$), and BCNI combined SWT (SWT)-treated rats ($n = 5$), western blots assessing the expression of TAZ (**k**) and Immunohistochemistry of TAZ (**l**). qRT-PCR (**m**) and immunohistochemistry (**n**) assessing the expression of *Adm* from the sham ($n = 12$), BCNI ($n = 8$), and BCNI combined SWT(SWT) treated rats ($n = 8$). Representative images (**o**) and quantifications (**p**) of the ICP from the WT (VED untreated, $n = 9$; VED treated, $n = 7$), *Yap* cKO (VED untreated, $n = 4$; VED treated, $n = 3$), Cre$^{AAV}$Y/T cKO (VED untreated, $n = 6$; VED treated, $n = 6$), Cre$^{Ad}$Y/T cKO (VED untreated, $n = 3$; VED treated, $n = 4$) mice treated with VED. **q** Distribution of different kinds of ED patients in VED therapy. The IIEF-5 score before and after the PDE5i non-responders, PDE5i side effects, PDE5i refusers or undefined with PDE5i ED patients treated with VED, *P* values were calculated by two-sides paired Student's *t* test, the confidence interval is 95%. Box plots indicate median (middle line), 25th, 75th percentile (box), minima, maxima and all points. Bar charts are presented as the mean ± sem. The statistical analysis was calculated by two-sides unpaired Student's *t* test (except **q**), the confidence interval is 95%. The point represents a mouse, rat or human biologically independent samples. Each experiment was repeated three independent times with similar results. Scale bars, 20 μm. Source data are provided as a Source Data file.

## Animals and genotyping

The following mice were used in this study: C57 BL/6, B6. FVB-Tg(Myh11-cre/ERT2)1Soff/J (Jackson lab: 019079), STOCK Wwtr1tm1Hmc Yap1tm1Hmc/WranJ (Jackson lab: 030532). Littermates that had Cre recombinase and were heterozygous for the floxed or knockout alleles were used as preferred controls where available. Animals were genotyped with standard procedures and with the recommended set of primers.

Approximately 250 g male SD rats and 6- to 8-week-old male mice were purchased from Soochow University Laboratory Animal Center (Suzhou, China). The animals were generated, housed, and bred under a 12 h light cycle at a temperature of $22 \pm 2\,^{\circ}C$, $55 \pm 5\%$ relative humidity, and with food and water ad libitum. Animals were maintained in the Animal Facilities of Soochow University under pathogen-free conditions. All studies involving mice and rats were approved by the Soochow University Institutional Animal Care and Use Committee. Tamoxifen was intraperitoneally injected for 7 consecutive days by 100 mg/kg body weight. Our study utilized both AAV-Cre and Adenovirus-Cre for in situ injection into the penis. The AAV-Cre was injected at a concentration of $5 \times 10^9$ PFU and ICP was measured after 3 weeks. The Adenovirus-Cre was administered once a week at a concentration of $1 \times 10^9$ PFU, with ICP measurements taken after 2 weeks. AAV-cre and Adenovirus-cre were obtained from Shanghai GeneChem Co.,Ltd.

## Cell culture

MOVAS cells (SMCs) were obtained from BeNa Culture Collection (BNCC338213) and cultured in DMEM (Gibco) supplemented with 10% fetal bovine serum (FBS), glutamine and antibiotics. HEK293T cells were obtained from ATCC (CRL-3216) and cultured in DMEM medium (Gibco) supplemented with 10% FBS and 1% penicillin/streptomycin. Primary penis corpus cavernosum smooth muscle cells were isolated from the penile tissue of mice and cultured in smooth muscle cell medium (ScienCell, Cat# 1101). The penis was removed and placed on ice, washed twice with PBS, cut into small tissue blocks with sterile surgical scissors, placed in small tissue blocks one by one at the bottom of the culture bottle, placed in the culture bottle upside down into the culture box for 4 h, and then added to the culture medium for 3-4 days for subculture. All cell lines were cultured, maintained and used within 10 passages according to the requirements.

## Atomic force microscopy (AFM) measurements

AFM analysis was performed to detect cell or penile tissue stiffness. The mice were treated with VED for 15 min, then we collected the flaccid or erectile penis and rapid freezed them using liquid nitrogen. Snap frozen tissue blocks were cut into 20 mm thick sections. Before measurements, each section was immersed in PBS, thawed at RT, and then maintained in proteinase inhibitor (Roche, Germany; cOmplete) in PBS. AFM indentations were analyzed using a Bruker mounted on an Olympus inverted microscope. Briefly, we used silicon nitride cantilevers with a spring constant of 0.03 N/m (Bruker, USA; MLCT) and attached a polystyrene spherical ball of 10 mm in diameter (Macklin, China) using epoxy glue (Pattex, China). Cantilevers were calibrated using the thermal oscillation method before each experiment. Five $10\,mm \times 10\,mm$ AFM force maps were typically obtained on each sample. The Hertz model was used to determine the elastic properties of the tissue. The upper 200 nm of tissue was considered for all fits. Tissue samples were assumed to be incompressible, and a Poisson's ratio of 0.5 was used in the calculation of the Young's elastic modulus.

## Bilateral cavernous nerve crush injury (BCNI) ED model

Approximately 250 g male SD rats or 6- to 8-week-old male mice were anesthetized with 40 mg/kg pentobarbital sodium. The cavernous nerve was identified posterolateral to the prostate. Bilateral cavernous nerves were crushed with Dumont forceps for 4 min each. All BCNI and sham surgeries were performed by a single surgeon for both rats and mice. For the sham surgery, cavernous nerves were not crushed, and the abdomens were closed after identifying the prostate and cavernous nerves.

## Prostate irradiation ED model

Approximately 250 g male SD rats were anesthetized with 40 mg/kg pentobarbital sodium and placed in the prone position. For the purpose of 3D images for treatment planning, a CT scan was performed by X-RAD SmART (Precision X-ray Inc., USA) to include the entire pelvis from the iliac crest to the tail base. The posterior lateral part of the prostate was selected and irradiated by a single fraction of 20 Gy X-ray.

## ICP measure

To measure ICP with cavernous nerve stimulation, animals were anesthetized by intraperitoneal injection of 40 mg/kg pentobarbital sodium. The right carotid artery was cannulated with polyethylene tubing containing heparinized saline for continuous monitoring of mean arterial pressure (MAP). The shaft of the penis was then exposed from skin and muscle and punctured with a 23-gauge needle connected to the tubing. Both were connected to a pressure transducer (Biopac system Inc, Goleta, CA, USA). The cavernous nerve was stimulated using a bipolar electrode connected to a stimulator. Stimulator settings were 5 V of 1.5 mA for 1 min with a minimum interval between stimulation of 5 min. The ICP/MAP ratio was determined using the maximum ICP divided by the MAP obtained during cavernous nerve stimulation. All data were recorded by an MP100 data acquisition system and analyzed using Acqknowledge software (Biopac System Inc., Goleta, CA, USA).

## Verteporfin

Mice were intraperitoneally injected with verteporfin 100 mg/kg in the dark. After injection, the needle was pulled out by rotation and

compressed for two minutes to prevent the outflow of drug solutions three times a week for two weeks. For established ED mouse models, verteporfin was injected for four weeks with the same treatment. The posterolateral prostate nerve of mice was stimulated by electricity savings, and the ICP value was measured to evaluate the erectile function of mice.

## Mouse castration model
Six- to eight-week-old male mice were anesthetized with 40 mg/kg pentobarbital sodium. The skin and pudendal part was cut from the lower edge of the penis. The testis was found on the basis of cutting the fat layer. After the vas deferens was ligated, both sides of the testis were removed. ICP value was measured after two weeks.

## Shock wave therapy
Approximately 250 g male SD rats were treated with BCNI to establish ED models, and then the ED rats were anesthetized with 40 mg/kg pentobarbital sodium. The ED rats were treated by shock wave physiotherapy twice a week for two weeks. The shock wave energy level was 5, the strike frequency was 300/time, and the strike interval was 1/0.5 s.

## VED
Approximately 250 g male SD rats or 6- to 8-week-old male mice were treated with BCNI or genetic deletion to establish ED models. The rats or mice were laid on the table in a supine position, and the penises were pumped through VED for 10 min at 600 mmHg 3 times per day. The treatment for ED rats last 28 days.

## RNA-seq
The mice were treated with VED for 15 min, then we collected the flaccid or erectile penis and rapid freezed them using liquid nitrogen. Total RNA was extracted using the Simply P Total RNA Extraction Kit (BioFlux) and prepared into a cDNA library according to the standard Illumina RNA-seq instructions. The generated cDNA library was sequenced in a $2 \times 150$ bp paired-end layout using an Illumina HiSeq2500. To estimate gene expression changes, the raw RNA-seq data were preprocessed using Trimmomatic to remove reads of low quality and adaptor contamination. The high-quality reads were further aligned to the mouse genome (mm10) using HISAT2 with default parameters. The mouse genome (mm10) was downloaded from the FTP of the Ensembl database. We retained only uniquely mapped reads for gene expression quantification at the count level using feature counts based on mouse gene annotation of Ensembl. The log2-transformed gene expression fold-changes (LFCs) and significantly differentially expressed (DE) genes were calculated.

## 2D mechanical challenging
For mechanical challenging of SMCs, cells were plated on standard fibronectin-coated tissue culture dishes or on fibronectin-coated hydrogels of the indicated stiffness (kPa). The hydrogels consisted of methylpropyl gelatin and photocatalyst LAP and were then irradiated with a 405 nm light source. The 0.25 kPa stiffness of the hydrogel was adjusted by controlling the time for 15 s. Meanwhile, the medium and high mechanics group were adjusted by controlling the time for 30 s and 5 min, respectively. Cells ($4 \times 10^4/cm^2$) were seeded in a drop of complete culture medium on top of fibronectin-coated hydrogels; after attachment, the hydrogel-containing wells were filled with the appropriate culture medium. Cells were collected for immunofluorescence after 48 h.

## 3-D collagen contraction assays
Collagen I solution (1.5 mg/mL) was prepared by neutralizing the pH of acid-solubilized rat tail collagen I (Cultrex) with NaOH and PBS buffer. SMCs or primary SMCs ($2 \times 10^5$ cells/well) were embedded in 200 μl/well neutralized collagen in 48-well ultralow attachment plates, the collagen hydrogel was polymerized at 37 °C for 1 h, and culture medium was added on top. After overnight incubation, the extent of contraction was assessed by subtracting the area of the gel from the well area using ImageJ software. Experiments were performed at least twice.

## RNA interference
siRNA transfections were performed with Lipofectamine RNAi-MAX (Thermo Fisher Scientific) in Opti-medium according to the manufacturer's instructions. Sequences of siRNAs are provided in Supplementary Table.

## Quantitative real-time PCR (qRT-PCR)
Total RNA from SMCs or primary SMCs was extracted using the Simply P Total RNA Extraction Kit (BioFlux), and contaminant DNA was removed by DNase treatment. Total RNA from the penises of rats was extracted using TRIzol reagent (Ambion, USA). qRT-PCR analyses were carried out on reverse-transcribed cDNAs with QuantStudio1 (Applied Biosystems, Thermo Fisher Scientific) and analyzed with QuantStudio Design & Analysis software (version 1.5.1). Expression levels are always normalized to GAPDH. PCR oligonucleotide sequences are listed in Supplementary Table.

## Western blots
Cells were lysed using lysis buffer (50 mM HEPES (pH 7.5), 100 mM NaCl, 50 mM KCl, 1% Triton X-100, 5% glycerol, 0.5% NP-40, 2 mM MgCl2, 1 μM DTT, cocktail and PMSF) followed by sonication and centrifugation at 4 °C. Extracts were quantified using the BCA method. Proteins were run on 4-12% SurePAGE-MOPS acrylamide gels (Genscript, China) and transferred onto PVDF membranes. Blots were blocked with 0.5% nonfat dry milk and incubated overnight at 4 °C with primary antibodies. Secondary antibodies were incubated for 1.5 h at room temperature, and then blots were developed with chemiluminescent reagents. Images were acquired with FluroChem M1 (Protein Simple). Uncropped scans were provided in the Source Data file.

The antibodies used for western blot were: anti-YAP/TAZ (Santa Cruz Biotechnology, sc-101199 (1:1000)), anti-YAP1 antibody (Proteintech, Cat#13584-1-AP (1:300)), anti-WWTR1 antibody (ATLAS, Cat#HPA007415 (1:300)) and anti-ADM antibody (Abcam, Cat#ab190819 (1:300)) and anti-GAPDH (Millipore, MAB374 (1:30000)). The secondary antibodies were from Beyotime Biotechnology (anti-mouse: Cat# A0216 (1:500), anti-rabbit: Cat# A0208 (1:500)).

## Immunofluorescence
Immunofluorescence on PFA-fixed cells and on PFA-fixed paraffin-embedded tissue slices was performed as previously described. Cells cultured on coverslips or penile samples were fixed with 4% paraformaldehyde for 10 min. The cultures were then washed 3 times with PBS, permeabilized with 0.5% Triton X-100 in PBS for 10 min, blocked with 10% BSA and 0.1% Triton X-100 in PBS for 1 h, incubated with primary antibodies (overnight, 4 °C), and then washed 3 times with PBS followed by incubation with secondary antibodies (1.5 h, room temperature). Phalloidin was added after the secondary antibodies for 15 min, whereas ProLong-DAPI was added after secondary antibodies or phalloidin for 15 min. Images were acquired using a confocal microscope (FV1200, Olympus). More than 5 fields of view from at least three independent experiments were randomly chosen.

Primary antibodies against YAP/TAZ (Santa Cruz Biotechnology Cat# sc-101199 (1:300)), phospho-MLC (Cell Signaling Technology Cat# 3671 (1:300)), αSMA (Abcam Cat# ab124964 (1:300)), PDGFRβ (Cell Signaling Technology Cat# 3169 (1:300)), CD31 (Cell Signaling Technology Cat# 77699 (1:300)), and phalloidin (PHDH1) were purchased from cytoskeleton. Secondary antibodies (1:300) were obtained from Beyotime Biotechnology (anti-mouse: Cat# A0460 and

A0428, anti-rabbit: Cat# A0423 and A0453). Samples were counter-stained with Prolong-DAPI (Molecular Probes, Life Technologies) to label cell nuclei.

## Immunohistochemistry and in situ hybridization

Penile tissues of mice or rats were embedded in OCT and then rapidly frozen. Cryostat sections were cut and dried on glass slides at room temperature. Then, cryostat sections were fixed with tissue fixation fluid for 30 min. Endogenous peroxidase blocking was performed by adding 1–2 drops of 3% hydrogen peroxidase, enough to cover the sections, followed by incubation for 10 min. The sections were blocked with 10% BSA in PBS for 1 h, incubated with primary antibodies (2 h, room temperature), washed 3 times with PBS, and incubated with secondary antibodies (0.5 h, room temperature). Immunoreactions were visualized using 3,3'-diaminobenzidine tetrahydrochloride hydrate (DAB) with subsequent counterstaining with Mayer's hema-toxylin using an inverted microscope (Leica). The antibodies were as follows: anti-YAP1 antibody (Proteintech, Cat# 13584-1-AP (1:300)), anti-WWTR1 antibody (ATLAS, Cat# HPA007415 (1:300)) and anti-ADM antibody (Abcam, Cat# ab190819 (1:300)).

The in situ hybridization was carried out according to methods provided from Boster (MK3908, China). In brief, nucleic acid frag-ments were exposed with 3% pepsin for 10 s. Adding *Cyr61* nucleotide hybridization solution and incubating at 4 °C overnight. After incu-bating biotinylated peroxidase for 2 h, immunoreactions were visua-lized using DAB with subsequent counterstaining with Mayer's hematoxylin using an inverted microscope (Leica).

## Flow cytometry

Smooth muscle cells were removed from the culture medium, washed twice with PBS, then incubated with 1 μM Fluo-4 Am (S1060, Beyotime) in 37 °C incubator for 30–60 min. Then the cells were washed once with PBS, digested with trypsin and collected, finally resuspended with 500ul PBS, and tested by flow cytometry (FACSVerse, BD, Z6511550318). Data were analyzed using FlowJo V10.8.1 (M11c3c353YH92SCS).

## Lentivirus preparation

HEK293T (ATCC) cells were transiently transfected with lentiviral vectors (10 μg per 60-cm² dish) together with packaging vectors pMD2-VSVG (2.5 μg) and pPAX2 (7.5 μg) using Lipofectamine 2000 (Invitrogen) according to the manufacturer's instructions.

## Chromatin immunoprecipitation (ChIP) coupled with quantita-tive real-time PCR (ChIP-qPCR)

ChIP-qPCR was performed as previously described. ChIP was typically performed in SMCs treated with Lat. A or sildenafil for 24 h. Briefly, cells were grown to a final concentration of $2 \times 10^7$ cells for each ChIP-qPCR experiment. Cells were crosslinked at room temperature by the addition of formaldehyde to a 1% final concentration for 10 min and quenched with 0.125 M final concentration of glycine. Crosslinked cells were scraped and resuspended in sonication buffer (50 mM HEPES-KOH pH 7.5, 140 mM NaCl, 1 mM EDTA, 1% Triton X-100, 0.1% Na-deoxycholate, 0.1% SDS) and sonicated using a Diagenode Bioruptor for three 5-minute rounds using pulsing settings (30 s ON; 30 s OFF). Sonicated chromatin was then incubated overnight at 4 °C with 1 μg of YAP/TAZ antibody conjugated to magnetic beads. Following IP, beads were washed twice with RIPA buffer (50 mM Tris-HCl pH 8, 150 mM NaCl, 2 mM EDTA, 1% NP-40, 0.1% Na-deocycholate, 0.1% SDS), low salt buffer (20 mM Tris pH 8.1, 150 mM NaCl, 2 mM EDTA, 1% Triton X-100, 0.1% SDS), high salt buffer (20 mM Tris pH 8.1, 500 mM NaCl, 2 mM EDTA, 1% Triton X-100, 0.1% SDS), LiCl buffer (10 mM Tris pH 8.1, 250 mM LiCl, 1 mM EDTA, 1% Na-deoxycholate, 1% NP-40), and 1X TE. Finally, DNA was extracted by reverse crosslinking at 65 °C for 6 h with proteinase K (20 μg/mL) and 1% SDS followed by DNA purification by a Qiagen PCR Purification Kit. After purification of the immunoprecipi-tated DNA, qRT-PCR was run at least in duplicate from at least two independent experiments, and data were normalized to input values and calculated as percent input recovery using the ΔΔCt method.

## Generation of single nucleus RNA sequencing (snRNAseq) libraries and sequencing

The rats were treated with VED for 10 min at 600 mmHg 3 times per day for 7 days. Then we collected the Con or VED penis and rapid freezed them using liquid nitrogen. Nuclei were extracted from rat penis using previous methods[41] and checked under microscope to investigate the morphology of nuclei. Extracted nuclei from two rat penis were mixed and subjected to single nucleus library construction using chromium single cell 3' reagent kits (v3 Chemistry) following the manufacture's protocol.

## Process of snRNAseq data

The raw sequencing data was mapped to the rat reference genome (Rnor_6.0, genome-date 2014-07, NCBI:GCA_000001895.4) using Cell Ranger (cellranger-6.1.2) with default parameters, giving rise to gene expression matrix for downstream analysis. The output of CellRanger was loaded into Seurat package[42]. Erection and flaccid data sets were integrated using Seurat IntegrateData function based on integration anchors identified by the FindIntegrationAnchors function. Top 4000 high variable features and top 20 principal components were selected for dimension reduction, followed with clustering using FindMarker function implemented in Seurat software at the resolution of 0.5. KEGG term enrichment was conducted using clusterProfiler[43] with the following parameters: organism = 'rno', keyType = 'kegg', pAdjust-Method = 'BH', minGSSize = 3, maxGSSize = 500. Data visualization was achieved using VlnPlot and DoHeatmap functions implemented in Seurat and other visualization functions embedded in complexheatmap[44] and ggplot[45]. Differentially expressed gene list between flaccid and erection states of SMC was extracted and sub-jected to transcription factor binding motif enrichment analysis using findMotifs.pl from HOMER[46] with the following parameters: input.txt rat Output/ -start −1000 -end 0 -len 4,10 -p 4.

## Preparation of the stretching chamber

Firstly, we made a mold of stretchable chamber and poured poly-dimethylsiloxane (PDMS) liquid (10:1 mixture of Dow Corning 184 gel and curing agent) into the mold. Then, we evacuated the air bubbles and baked the PDMS in the mold at 80 °C for 2 h. Afterward, the PDMS stretching chamber for experiment was obtained by disassembling the mold. Then the surface of the chamber was coated with a 0.2 mg/mL collagen solution at 37 °C for 24 h. Finally, the stretching chamber was rinsed with water, air dried, and sterilized by UV light for 10 min before seeding the cells.

## Stretching method

The stretching chamber was sleeved on the two arms of the stretching device. Place the entire stretching device on the microscope stage which itself was in the living cell workstation. The temperature in the workstation was kept at 37 °C for long-term in-suit observation. To keep a stable concentration of 5% carbon dioxide for the cells, a flexible film was used to cover the stretching chamber to build an approximately closed space, and pass 5% carbon dioxide and water vapor into the closed space. Through a screw with opposite threads, the rotary motion of the stepper motor was translated into a symmetrical linear motion of the stretching arms of the device. The flexible chamber fixed on the stretching arms was then stretched by the periodic motion of the arms with different amplitudes at specific frequencies. In the experiment, we chose the stretching amplitudes as 10% of the

length of the stretching chamber, and the stretching frequency as 1.5 Hz. The loading mode is a stretch-recovery-stretch cycle, which was performed for 6 h in each experiment.

## Statistics

The number of biological and technical replicates and the number of animals are indicated in the figure legends, main text and Methods. In addition, all data are presented as the mean ± sem. in the figure legends and extended data figures. Statistical analysis was performed by excel software. Comparisons were analyzed by two-sides unpaired Student's t test, as indicated in the figure legends and extended data figures. The investigators were blinded to allocation during experiments and outcome assessment.

## Reporting summary

Further information on research design is available in the Nature Portfolio Reporting Summary linked to this article.

## Data availability

Single nucleus RNA sequencing data from rats and RNA sequencing data from mice can be accessed from the NCBI Gene Expression Omnibus database GSE208293. The mouse genome (mm10) was downloaded from the FTP of the Ensembl database (https://asia.ensembl.org/Mus_musculus/Info/Index). All other relevant data supporting the key findings of this study are available within the article and its Supplementary Information files. Source data are provided with this paper.

## Code availability

All code associated with this manuscript have been uploaded to GitHub (https://github.com/jmt1105/ED).

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

## Acknowledgements

The authors thank Zhifang Chai, Zhenke Wen, Yiran Zheng and Tom. K. Hei for helpful discussions. This work was supported by grants from National Natural Science Foundation of China (31971165 and 82173465, L.C.); National Natural Science Major Project (82192883, G.M.-Z.); The Fok Ying-Tong Education Foundation of China (171017, L.C.); Leading Talents Program of Gusu District (ZXL2022454, L.C.); CAMS Innovation Fund for Medical Sciences (CIFMS) (2021-I2M-1-061, D.S.-C.).

## Author contributions

Conceptualization: M.T.-J., Y.D.-X., D.S.-C., and L.C. Methodology: M.T.-J., D.S.-C., Y.Y.-S., Z.S.-Z., S.D., L.J.-Z., H.M.-Z., X.N.-J., Y.L., Y.F.-Z., S.Y.-W., W.S.-Z. Investigation: M.T.-J., Y.D.-X., Y.Y.-S., S.D. Visualization: M.T.-J., D.S.-C., Y.D.-X., Y.F.-Z., Y.Y., and L.C. Funding acquisition: L.C. Project administration: M.T.-J., D.S.-C., Y.D.-X., B.H.-J., G.M.-Z., Y.Y., and L.C. Supervision: Y.Y. and L.C. Writing—original draft: M.T.-J., B.Y.-L., and L.C. Writing—review & editing: Y.D.-X., D.S.-C., Y.Y., and L.C.

## Competing interests

The authors declare no competing interests.

## Additional information

[1]State Key Laboratory of Radiation Medicine and Protection, School of Radiation Medicine and Protection, Collaborative Innovation Center of Radiation Medicine of Jiangsu Higher Education Institutions, Medical College of Soochow University, 215123 Suzhou, China. [2]Institute of Systems Medicine, Chinese Academy of Medical Sciences & Peking Union Medical College, and Suzhou Institute of Systems Medicine, 215123 Suzhou, China. [3]Department of Urology, The Affiliated Hospital of Changchun University of Chinese Medicine, 130021 Changchun, China. [4]Institute of Biomechanics and Applications, Department of Engineering Mechanics, Zhejiang University, 310027 Hangzhou, China. [5]Wenzhou Institute, University of Chinese Academy of Sciences, 325001 Wenzhou, China. [6]Binzhou Medical University, 264003 Yantai, China. [7]Department of Nutrition and Food Hygiene, Soochow University of Public Health, 215123 Suzhou, China. [8]Department of Urology, Beijing Friendship Hospital, Capital Medical University, 100050 Beijing, China. [9]These authors contributed equally: Mintao Ji, Dongsheng Chen. ✉e-mail: yangyong@ccucm.edu.cn; xyongd@yeah.net; changlei@suda.edu.cn

