## [Peer Review File · Nature Communications]

The Role of Mechanoregulated YAP/TAZ in Erectile Dysfunction.REVIEWER COMMENTS

Reviewer #1 (Remarks to the Author):

In this study, authors provide abundant experimental evidence to prove that YAP/TAZ-Adrenomedullin cascade improved erectile function even in PDE5i non-responders. However, there are still some questions that may affect the proof of the conclusion need to answer.

1. FIG1 a-c. NPT is a commonly used clinical test to assess erectile function. I am not sure why the authors can conclude this hypothesis "These clinical observations raise the question of whether sporadic stretching of the 64 penis and its associated mechanical stimulation is necessary to maintain the function of penile erections."

2. Using three species may introduce additional interference, and the authors used human clinical samples and mouse models in Fig1 and rats in single-cell RNA-seq studies. Whether the differences in penile anatomy, erectile mechanism, and ED pathogenesis among the three species will affect the experimental conclusions

3. Another question is that the authors need to explain in more detail how RNA sequencing be performed in erect or flaccid states. I am concerned about whether significant changes in transcript levels can be made in such a short time. In the process of single-cell digestion, the cells have lost their mechanical force. Will this interfere with the sequencing results? For example, the absence of the HIPPO pathway in Fig1j may be a proof.

4. Interestingly, previous studies have shown that in humans, The nuclear YAP and CTGF levels of ED patients (no response to PDE5i) are usually higher than those of normal controls (Zhao, L., et al. (2022). "Single-cell transcriptome atlas of the human corpus cavernosum." *Nat Commun* 13(1): 4302.), which seems to contradict the results in FIG. 2. Is this difference caused by species differences, or does it indicate that both YAP deletion or overactivation are not conducive to the repair of cavernosal tissue? In addition, CTGF is usually upregulated in muscular dystrophy or muscular dystrophy, "Role of Matricellular CCN Proteins in Skeletal Muscle: Skeletal Muscle; Focus on CCN2/CTGF and Its Regulation by Vasoactive Peptides." *Int J Mol Sci* 22(10).) However, it seems to be a protective factor in this study and the authors need to discuss it in more detail.

5. Although the authors demonstrate VED and SWT simulate mechano-yap/Taz-adm axis in a mouse model, VED and SWT may have a more complex mechanism in patients, whether CC tissue or CC blood can be taken from the patient for verification (such as CTGF or ADM), if not, I suggest that this section could be omitted.

6. The therapeutic effects of VED and SWT have been widely accepted in clinic, but they also have limitations. but it also has limitations for some patients. Authors showed that VED and SWT upregulating YAP, does this suggest that VED may not be effective until the cause of YAP downregulation (such as

aging or diabetes) is not treatable.

7. Is the YAP pathway unaffected in ED patients who are PDE5I responsive? Since different

8. Does the YAP pathway remain unaffected in patients who respond to PDE5I? All disease models in this paper lead to YAP downregulation, which again seems to be a common phenomenon. Which etiologies of ED are associated with YAP and TAZ, and which are not?

9. Some false or confusing descriptions, such as in Materials and Methods (LINE 557) author used rat penis for single-cell sequencing, but in Results section the authors used mice (LINE 79).

Reviewer #2 (Remarks to the Author):

In the present study the authors identified YAP/TAZ activation by stretching of SMCs as a critical factor in sustaining an erection. Nuclear YAP/TAZ directly regulates the transcription of Adrenomedullin (ADM), a locally acting hormone controlling vascular tone. Diffusing ADM promotes smooth muscle relaxation to sustain an erection. This is an interesting article documenting the role of YAP/TAZ and identifies its underlying molecular mechanism as a mechano-regulated YAP/TAZ-ADM molecular axis. However, there are substantial issues that should be clarified

1. The authors used the BCNI ED model as main animal models, however, as we know, vasculogenic ED accounts for about 70% of ED and neuropathic ED about 20-30%. Therefore, what is the role of YAP/TAZ signaling in vasculogenic ED model, such as diabetic ED model. Even though this study is focused on the BCNI ED, it would be better to add some discussion of YAP/TAZ in vascular regeneration, and even some ideal hypotheses for YAP/TAZ in vasculogenic ED, because YAP/TAZ is also well known in vascular regeneration. If possible, for further study, the YAP/TAZ function in diabetic ED will be more interesting and more useful for ED treatment.

2. Penile erection requires well-coordinated interaction between endothelial cells, pericyte, smooth muscle cells, and neuronal cells. Recently much attention has focused on the role of endothelial cells and pericytes. However, this study has somewhat limited values because the authors only focused on the smooth muscle cells. Therefore, additional studies are needed to elucidate the role of YAP/TAZ pathway in other cell types, such as endothelial cells and pericytes.

3. In the abstract section, the authors described that conventional PDE5i targeting NO-cGMP signaling does not cure YAP/TAZ deficient ED. In contrast, by activating YAP/TAZ-Adrenomedullin cascade, mechano-stimulation improved erectile function, including PDE5i non-responders in both experimental models and clinical data.

In animal experiment, the authors have treated tadalafil in YAP/TAZ deficient ED mice and they defined YAP/TAZ deficient ED mice as a PDE5i non-response animal model. And then they insisted that activating

YAP/TAZ-Adrenomedullin cascade, mechano-stimulation improved erectile function in PDE5i non-response animals (YAP/TAZ deficient ED mice). I think this is not a logical approach. The use of specific activators of YAP/TAZ in animal models of ED which are proven to non-responder to PDE5 inhibitors, such as type I/II diabetes, dyslipidemia, and cavernous nerve injury, is needed to test whether those activators rescue erectile function to reach the conclusion. For the adrenomedullin, previous study in an animal model of diabetic ED reported that adenovirus expressing adrenomedullin induced only partial improvement of erectile function (J Sex Med, 2013;10:1707-19).

Moreover, in human study, the authors reported that the curative effect of VED in PDE5i non-response ED patients was significant. However, both PDE5 inhibitors and VED are the first-line treatment modalities for ED. The results of VED cannot justify the activation of YAP/TAZ recover erectile function in men with PDE5 inhibitor-non-responders.

Reviewer #3 (Remarks to the Author):

Ji et al present an interesting and mostly rigorous set of findings describing the mechano-responsive activity of YAP/TAZ in penile SMCs to induce expression of adrenomedullin to facilitate penile erection.

Strengths of the manuscript include a comprehensive and mechanistic study of these molecules and the predominant triggers and physiologic responses that are linked to this mechanism of penile erection. Study numbers, statistical analysis, and logic all score highly.

There were some deficiencies in experimentation and logic, however, that should be rectified.

1. The authors make a strong claim that YAP/TAZ activation in this setting is mechano-sensitive. This makes sense given the large number of papers that have described this type of action for these molecules. However, the type of stretch that is seen by penile SMCs may be very specific and different than in other bodily contexts. Here, the authors offer a confusing description of the platform for studying stretch in vitro and may be suboptimal. As far as I can tell, they are using matrix that is either stretched or non-stretched. Yet, they do not physically stretch the cells themselves, which in my opinion should be the predominant way in which these SMC would see mechanoactivation. Thus, I would suggest that the authors include direct stretch exposure to determine if that also is inducing YAP/TAZ. If, on the other hand, the authors believe the stretched matrix is truly the cause, then I would argue that there may be differences in the matrix composition (protein, metabolites, etc.) that could be activating YAP/TAZ and not necessarily stretch of the SMCs itself. If so, the authors should attempt to quantify and characterize the stretch vs. non-stretch matrix to determine is the composition or the physical properties of the matrix the trigger for YAP/TAZ activation.

2. The in vivo mouse model that carries MYH11-Cre and is tamoxifen sensitive should result in KO of YAP/TAZ in all smooth muscle cells and not just penile SMCs. However, a confounder of that model is that such treatment likely affects resting blood pressure (as vasodilation should increase in multiple

vascular beds) and could compromise penile erection due to poor blood flow in general (rather than specifically affecting YAP/TAZ activity in the penile tissue). The authors should determine vasodilatory state in the mouse before and after such systemic injection of tamoxifen and global SMC KO. To rectify, the authors may need to perform the same experiment with direct KO only in penile tissue (i.e., AAV-Cre injected into penile tissue of YAP/TAZ f/f mice).

3. In that vein, a powerful in vivo experiment that is missing is to use such a penile-specific KO of YAP/TAZ to show that VED treatment is not effective in these mice.

4. The authors should comment on whether the use of YAP/TAZ inhibitors that are discussed for cancer may be an issue with erectile dysfunction if they are administered systemically. Since the YAP inhibitor verteporfin is used as a FDA approved drug for ophthalmologic disease, is there any evidence that such patients suffer from erectile dysfunction?

Minor:

-- English and grammar may need to be improved. For example, Fig. 3 title "Mechanically stretches dominate..." does not make sense.

POINT TO POINT TO ANSWER REVIEWER COMMENTS: (Responses are in black.)

Reviewer #1 (Remarks to the Author):

In this study, authors provide abundant experimental evidence to prove that YAP/TAZ-Adrenomedullin cascade improved erectile function even in PDE5i non-responders.

Response: Thank you for your supportive feedback.

However, there are still some questions that may affect the proof of the conclusion need to answer.

1.FIG1 a-c. NPT is a commonly used clinical test to assess erectile function. I am not sure why the authors can conclude this hypothesis "These clinical observations raise the question of whether sporadic stretching of the 64 penis and its associated mechanical stimulation is necessary to maintain the function of penile erections."

Response: We are sorry for any confusion. The nocturnal penile tumescence (NPT) test is commonly used in clinical evaluations of erectile function as it measures the physiological phenomenon of several erections during sleep in healthy males. Conversely, the lack of nocturnal erections is a hallmark of erectile dysfunction (ED). Given that physiological phenomena serve biological functions, we posit that regular penile erections in healthy males may be indicative of changes in penile tissue stiffness. Conversely, the absence of mechanical stimulation in daily life may contribute to ED. Our hypothesis is that mechanical stimulation plays a crucial role in maintaining erectile function, and this has been further explored in subsequent experiments. We apologize for any confusion and have revised the description in the text (lines 66-73).

2. Using three species may introduce additional interference, and the authors used human clinical samples and mouse models in Fig1 and rats in single-cell RNA-seq studies. Whether the differences in penile anatomy, erectile mechanism, and ED pathogenesis among the three species will affect the experimental conclusions

Response: We believe this is a common challenge in the examination of human diseases. Which is balancing ethical considerations with the need for accurate results. Animal models are often used to gain a deeper understanding of disease mechanisms. In this study, we used both rats and mice as animal models to gain insights into the mechanisms of erectile dysfunction, meanwhile, rats/mice as the ED model due to their widespread use in previous ED studies (as reviewed in PMID: 18558990, 21981717 and 25624570). In addition to the ethical and practical advantages, the use of mice in our study also enabled precise genetic modification, facilitating the identification of specific genes involved in the development of ED. Specifically, we generated mice with SMC-specific and penis-specific (by locally injecting a Cre-expressing virus into the penis of *Yap^{fl/fl};Taz^{fl/fl}* mice) YAP/TAZ knockouts, which exhibited ED symptoms (Fig 2a and 6o-p, columns 5 and 7 in comparison with control columns 1). Our findings using these tissue-specific YAP/TAZ knockout mice strongly support the essential role of these proteins in maintaining penile erectile function. To validate our findings in a real-world scenario, we also

examined YAP/TAZ expression in human samples from healthy males and ED patients (Fig. 2c). Information was also collected from patients undergoing standard VED treatment from a collaborating hospital to test if mechanical therapy can cure PDE5i non-responders (Fig. 6q). And most importantly, all three different species share the same mechanism, that is, the mechano-YAP/TAZ-Adrenomedullin cascade improves erectile function even in PDE5i non-responders.

3. Another question is that the authors need to explain in more detail how RNA sequencing be performed in erect or flaccid states. I am concerned about whether significant changes in transcript levels can be made in such a short time. In the process of single-cell digestion, the cells have lost their mechanical force. Will this interfere with the sequencing results? For example, the absence of the HIPPO pathway in Fig1j may be a proof.

Response: We agree with your perspective that the digestion process of cells may have resulted in diminished mechanical force and thus low YAP/TAZ activity, as demonstrated in a previous study (PMID: 22215811), which showed that cell detachment inhibits YAP/TAZ activation. To mitigate this, we employed a strategy of rapid freezing using liquid nitrogen to preserve the cellular organization, followed by single-nucleus RNA-Seq by nuclear detection, which reduced the impact of environmental variations. Nuclear extraction instead could keep the mechanic impact on YAP/TAZ, as evidenced by multiple studies to separate nuclear YAP under various mechanical conditions (PMID: 31243273, 30401838, 35246511, 35562016 and 30135582). In order to imitate the normal physiological state of penile tumescence (NPT), we utilized a VED to induce three rounds of penile erection, collected samples during the erect state, and documented the methodology in Line 606-607. Additionally, Fig. 1m displays a significant increase in the in situ hybridization (ISH) assay of Cyr61, a YAP/TAZ target gene, in the erect penis compared to the flaccid penis, demonstrating the successful capture of cells responsive to mechanical force.

Furthermore, in response to your recommendation, we have re-analyzed the single-nuclear RNA-seq data and identified changes in the Hippo pathway in multiple type of cells, including smooth muscle cells (SMCs). A heatmap analysis of YAP/TAZ target genes revealed a significant elevation of YAP/TAZ activity in the erect penis in comparison to the flaccid penis. These findings have been incorporated into Fig. 1i-l and the manuscript has been revised in lines 84-101.

4. Interestingly, previous studies have shown that in humans, The nuclear YAP and CTGF levels of ED patients (no response to PDE5i) are usually higher than those of normal controls (Zhao, L., et al. (2022). "Single-cell transcriptome atlas of the human corpus cavernosum." *Nat Commun* 13(1): 4302.), which seems to contradict the results in FIG. 2. Is this difference caused by species differences, or does it indicate that both YAP deletion or overactivation are not conducive to the repair of cavernosal tissue? In addition, CTGF is usually upregulated in muscular dystrophy or muscular dystrophy, "Role of Matricellular CCN Proteins in Skeletal Muscle: Skeletal Muscle; Focus on CCN2/CTGF and Its Regulation by Vasoactive Peptides." *Int J Mol Sci* 22(10).) However, it seems to be a protective factor in this study and the authors

need to discuss it in more detail.

Response: Thank you for bringing the paper (PMID: 35879305) to our attention. Our findings contrast with those presented in the paper. This may be due to the fact that the paper's primary method, single-cell RNA-seq from digestion to isolated cells, has the potential to miss important mechanical information during sample preparation, as you previously noted. If all studies rely on the same method that may miss mechanical information, it is not surprising that their conclusions differ from ours. Additionally, we believe that bioinformatic conclusions must be validated through experiments. Their paper evaluated YAP/TAZ activity in vascular-related and non-vascular related ED through limited experiments on YAP activity and YAP immunofluorescence staining in dozens of isolated endothelium cells (more specifically, 30 isolated cells!). Our research provides more comprehensive and extensive evidence to support the role of YAP/TAZ in penile erection. We utilized single-nucleus RNA-seq, which preserved the mechanical impact on cells, to discover the essential role of YAP/TAZ in penile erection (Fig. 1i-l). Further evidence was provided by the demonstration of YAP/TAZ activity loss in multiple ED models in rats and mice, such as BCNI, Radiation, and Castration (Fig. 2d-m). This was confirmed by tissue-specific YAP/TAZ knockout mice that induced ED (Fig. 2a). Lastly, analysis of YAP/TAZ activity from multiple clinical ED patients reinforced the conclusion that YAP/TAZ activity is necessary for penile erection (Fig. 2c).

Regarding CTGF (Ccn2), in our study it was used as a marker of YAP/TAZ activity, along with Cyr61, due to its role in transcriptional regulation. Both Ctgf and Cyr61 are well-established markers of YAP/TAZ transcriptional activity in the Hippo pathway study (as referenced PMID: 30401838, 35768505, 21654799, 24658687 and 23954413). However, YAP/TAZ may not necessarily perform its role through Ctgf. Our further investigation found that Adm, a target of YAP/TAZ, plays an important role in ED. Additionally, skeletal muscle was different from smooth muscle cells, and the role of Ccn in skeletal muscle is not a focus of our research. To clarify this, we have discussed and revised.

5. Although the authors demonstrate VED and SWT simulate mechano-yap/Taz-adm axis in a mouse model, VED and SWT may have a more complex mechanism in patients, whether CC tissue or CC blood can be taken from the patient for verification (such as CTGF or ADM), if not, I suggest that this section could be omitted.

Response: Thanks for your suggestion. We also believe it is important. However, due to ethical considerations, it is challenging to conduct such studies.

6. The therapeutic effects of VED and SWT have been widely accepted in clinic, but they also have limitations. but it also has limitations for some patients. Authors showed that VED and SWT upregulating YAP, does this suggest that VED may not be effective until the cause of YAP downregulation (such as aging or diabetes) is not treatable.

Response: You are right, and Rev3 shared the same opinion. Our proposed mechanism suggests that external mechanical stimulation using VED and SWT can effectively treat PDE5i non-

responders by increasing the YAP/TAZ-ADM signaling pathway. However, in cases of prior loss of YAP/TAZ activity, such as in the case of specific knockout of YAP/TAZ from mice's penis, additional mechanical stimulation (in this instance, VED) will not produce a therapeutic effect. This finding is supported by our newest experimental data presented in Fig. 6o-p, which shows that VED has no curative effect on either SMC-specific or penile-specific YAP/TAZ-deficient ED mice.

7. Is the YAP pathway unaffected in ED patients who are PDE5I responsive? Since different

Response: This question appears to be similar to the one in question 8. We kindly direct you to our response in question 8 for further clarification.

8. Does the YAP pathway remain unaffected in patients who respond to PDE5I? All disease models in this paper lead to YAP downregulation, which again seems to be a common phenomenon. Which etiologies of ED are associated with YAP and TAZ, and which are not?

Response: Penile erection is triggered by smooth muscle cells (SMCs) relaxation induced by NO-cGMP activation. In this study, we found that an erection causes stretching of the penile SMCs, which activates YAP/TAZ. Nuclear YAP/TAZ directly controls the transcription of Adrenomedullin (ADM), a locally acting hormone that regulates blood vessel tone. ADM diffuses and sustains intracellular cAMP in SMCs, leading to feedback on NO release and smooth muscle relaxation that sustains an erection. Our research uncovered a secondary layer of regulation for erections beyond the traditional NO-cGMP pathway, the "mechano-YAP/TAZ-ADM" axis, which is activated after NO-cGMP stimulation and regulates SMC relaxation through the modulation of ADM transcription and calcium levels (Fig. 5j). Adding mechanical stimuli to activate this axis should have therapeutic effects on both PDE5i responders and non-responders, as supported by clinical data on the efficacy of vacuum erection devices in curing ED in both groups (Fig. 6q). It is important to note that deactivating YAP/TAZ in mice leads to a form of ED that does not respond to treatment with PDE5 inhibitors or mechanical stimuli, as shown by conditional knockout experiments (Fig. 3u and 6o-p). This suggests the critical role of YAP/TAZ in erection control. Nonetheless, we have to acknowledge the limitations of the study, as it did not examine all forms of ED, such as vasculogenic ED. Meanwhile, with limited sequencing data from ED patients, it is currently challenging to determine which etiologies of ED are associated with decreased YAP/TAZ activity. We have included a discussion of these limitations in the text and have suggested that further research is necessary to determine the universal applicability of the proposed mechanism and gain a deeper understanding of the relationship between YAP/TAZ activity and ED (for example in vasculogenic ED).

9. Some false or confusing descriptions, such as in Materials and Methods (LINE 557) author used rat penis for single-cell sequencing, but in Results section the authors used mice (LINE 79).

Response: Thanks, corrected.

Reviewer #2 (Remarks to the Author):

In the present study the authors identified YAP/TAZ activation by stretching of SMCs as a critical factor in sustaining an erection. Nuclear YAP/TAZ directly regulates the transcription of Adrenomedullin (ADM), a locally acting hormone controlling vascular tone. Diffusing ADM promotes smooth muscle relaxation to sustain an erection. This is an interesting article documenting the role of YAP/TAZ and identifies its underlying molecular mechanism as a mechano-regulated YAP/TAZ-ADM molecular axis.

Response: Thanks for your comments, and perfectly summarized the MS.

However, there are substantial issues that should be clarified

1. The authors used the BCNI ED model as main animal models, however, as we know, vasculogenic ED accounts for about 70% of ED and neuropathic ED about 20-30%. Therefore, what is the role of YAP/TAZ signaling in vasculogenic ED model, such as diabetic ED model. Even though this study is focused on the BCNI ED, it would be better to add some discussion of YAP/TAZ in vascular regeneration, and even some ideal hypotheses for YAP/TAZ in vasculogenic ED, because YAP/TAZ is also well known in vascular regeneration. If possible, for further study, the YAP/TAZ function in diabetic ED will be more interesting and more useful for ED treatment.

Response: We concur that examining the impact of the mechano-YAP/TAZ-ADM axis on vascular regeneration is intriguing. Our single-nucleus RNA sequencing data analysis of gene profiles in endothelial cells found significant changes in the Hippo pathway and YAP/TAZ activity during an erection (Fig. S8b). However, we observed a negative signal for YAP/TAZ immunofluorescence in penile sections. Additionally, we found that the endothelial cells remain unaltered during YAP/TAZ and ADM activation (Fig. S8h). These results suggest that YAP/TAZ responds to mechanical stimuli in endothelial cells. Nonetheless, in our experiment, within the window of detecting erection function restoration, the effect of YAP/TAZ-ADM on endothelial cells was limited. On the other, we have an intriguing observation of the changes in YAP/TAZ activity in pericytes during ED progression. Our results also demonstrate a correlation between YAP/TAZ activity in pericytes and ED progression following BCNI treatment in rats. Interestingly, mechanical stimuli treatment was found to significantly enhance YAP/TAZ activity in pericytes compared to non-treated BCNI-ED rats, providing evidence of the presence of a mechano-YAP/TAZ cascade in pericytes (Fig. S8e-g). Given the distinct responses of pericytes and endothelial cells, we postulate that the mechano-YAP/TAZ could have a role in vascular system.

We selected the BCNI ED model because it is known to have a lower response to PDE5i, as evidenced in both literature (PMID: 28296277 and 21839578) and our own experiment (Fig. 3v and S6a). The challenge of investigating vasculogenic ED is due to the difficulties and time required to create animal models that are non-responsive to PDE5i. Despite these, we have thoroughly addressed the limitations of this study within the text (line 368-380).

2. Penile erection requires well-coordinated interaction between endothelial cells, pericyte, smooth muscle cells, and neuronal cells. Recently much attention has focused on the role of endothelial cells and pericytes. However, this study has somewhat limited values because the authors only focused on the smooth muscle cells. Therefore, additional studies are needed to elucidate the role of YAP/TAZ pathway in other cell types, such as endothelial cells and pericytes.

Response: Thanks for your advice. Following your suggestion, we performed single-nucleus RNA sequencing analysis on gene profiles in both endothelial cells and pericytes during an erection. Our findings showed changes in the Hippo pathway and YAP/TAZ activity in both cell types (Fig. S8a-b), but negative signals of YAP/TAZ in endothelial cells led us to focus on pericytes. We observed changes in YAP/TAZ activity in pericytes during ED progression, with a decrease in nuclear-localized YAP/TAZ after BCNI treatment, reaching the lowest point on day 14 and being completely restored on day 60 (Fig. S8c-d). Our results show that YAP/TAZ activity in pericytes is also closely linked to ED progression after BCNI treatment. Furthermore, mechanical stimuli treatment significantly increased YAP/TAZ in pericytes compared to non-treated BCNI-ED rats (Fig. S8e-g), suggesting that the mechano-YAP/TAZ cascade is also active in pericytes.

3. In the abstract section, the authors described that conventional PDE5i targeting NO-cGMP signaling does not cure YAP/TAZ deficient ED. In contrast, by activating YAP/TAZ-Adrenomedullin cascade, mechano-stimulation improved erectile function, including PDE5i non-responders in both experimental models and clinical data.

In animal experiment, the authors have treated tadalafil in YAP/TAZ deficient ED mice and they defined YAP/TAZ deficient ED mice as a PDE5i non-response animal model. And then they insisted that activating YAP/TAZ-Adrenomedullin cascade, mechano-stimulation improved erectile function in PDE5i non-response animals (YAP/TAZ deficient ED mice). I think this is not a logical approach. The use of specific activators of YAP/TAZ in animal models of ED which are proven to non-responder to PDE5 inhibitors, such as type I/II diabetes, dyslipidemia, and cavernous nerve injury, is needed to test whether those activators rescue erectile function to reach the conclusion. For the adrenomedullin, previous study in an animal model of diabetic ED reported that adenovirus expressing adrenomedullin induced only partial improvement of erectile function (J Sex Med, 2013;10:1707-19).

Moreover, in human study, the authors reported that the curative effect of VED in PDE5i non-response ED patients was significant. However, both PDE5 inhibitors and VED are the first-line treatment modalities for ED. The results of VED cannot justify the activation of YAP/TAZ recover erectile function in men with PDE5 inhibitor-non-responders.

Response: We did follow your advice, and now show that activation of YAP/TAZ with the specific activator PY60 can rescue erectile function in ED animal models that do not respond to PDE5 inhibitors. In the BCNI-induced ED mice, which showed no response to PDE5i (Fig.3v, compared in columns 2 and 3), treatment with PY60 significantly rescued erectile function (Fig. 3v, compared in columns 2 and 4). These results provide further evidence for our conclusion.

We would like to clarify that when we mention "PDE5i non-response animals," we don't imply that they are solely "YAP/TAZ deficient ED mice." There can be several causes, such as type I/II diabetes, dyslipidemia, and cavernous nerve injury, that result in PDE5i non-response. Our study revealed that an erection trigger stretching of penile SMCs, activating YAP/TAZ. Nuclear YAP/TAZ then directly regulates the transcription of Adrenomedullin (ADM), which sustains intracellular cGMP levels in SMCs and leads to feedback-mediated NO release and smooth muscle relaxation, thereby sustaining an erection (Fig. 5j). We see that you and Rev3 share the same concerns. If our model is accurate, the YAP/TAZ-ADM cascade should mediate the effects of mechanical stimuli on erection. Therefore, extra mechanical stimuli should not be curative in YAP/TAZ deficient ED mice. Our new data in Fig. 6o-p confirms that VED has no curative effect on either SMC-specific or penile-specific YAP/TAZ deficient ED mice (as established by local injection of Cre-expressing virus). Meanwhile, ADM acts as a downstream target of YAP/TAZ and that supplementation of ADM in YAP/TAZ deficient ED mice can partially rescue erectile function. This contrasts with the observed no effect with PDE5i treatment (Fig.5h and Fig.3u). We would like to acknowledge the limitations of our study in that we did not investigate our mechanism in vasculogenic ED. Nonetheless, we are encouraged by published studies suggesting that ADM may partially improve vasculogenic ED through this mechanism. We appreciate your suggestion and have included this information in the discussion section of our paper.

In human study, it is also suggested by Rev1 in Question 5. We believe it is important. However, ethical considerations and with time limitation make it difficult to carry out such studies, and we must find a balance between ethical considerations and the need for accurate results.

Reviewer #3 (Remarks to the Author):

Ji et al present an interesting and mostly rigorous set of findings describing the mechano-responsive activity of YAP/TAZ in penile SMCs to induce expression of adrenomedullin to facilitate penile erection.

Strengths of the manuscript include a comprehensive and mechanistic study of these molecules and the predominant triggers and physiologic responses that are linked to this mechanism of penile erection. Study numbers, statistical analysis, and logic all score highly.

Response: We are grateful for the reviewer's positive feedback on our story. In this revision, we have made significant efforts to reinforce the central findings of the paper in accordance with the reviewer's suggestions. Thank you.

There were some deficiencies in experimentation and logic, however, that should be rectified. 1. The authors make a strong claim that YAP/TAZ activation in this setting is mechano-sensitive. This makes sense given the large number of papers that have described this type of action for these molecules. However, the type of stretch that is seen by penile SMCs may be very specific and different than in other bodily contexts. Here, the authors offer a confusing

description of the platform for studying stretch in vitro and may be suboptimal. As far as I can tell, they are using matrix that is either stretched or non-stretched. Yet, they do not physically stretch the cells themselves, which in my opinion should be the predominant way in which these SMC would see mechanoactivation. Thus, I would suggest that the authors include direct stretch exposure to determine if that also is inducing YAP/TAZ. If, on the other hand, the authors believe the stretched matrix is truly the cause, then I would argue that there may be differences in the matrix composition (protein, metabolites, etc.) that could be activating YAP/TAZ and not necessarily stretch of the SMCs itself. If so, the authors should attempt to quantify and characterize the stretch vs. non-stretch matrix to determine is the composition or the physical properties of the matrix the trigger for YAP/TAZ activation.

Response: Thank you for your suggestion. To reinforce our core findings, we performed experiments where we stretched smooth muscle cells (SMCs) on stretching system. Our results showed that highly stretched SMCs exhibited elevated YAP/TAZ activity, regardless of the presence or absence of PDE5i (Fig. 3e-h). Following the pioneering work of Stefano Piccolo's lab on mechanotransduction of YAP/TAZ (as referenced PMID: 23954413), we also applied various mechanical forces to the cells, including stretching, variations in cell density, and different stiffness of extracellular matrices (ECMs). By incorporating multiple experimental strategies, we aim to demonstrate the robustness of our results. Our findings showed that cells subjected to high mechanical input (stretching, sparse, and stiff ECM) resulted in elevated YAP/TAZ activity, with the protein located in the nucleus and transcriptional activity insensitive to PDE5i (Fig. 3e-j and S3c-f). Our experiments utilized Methacrylate gelatin (GelMA) as the extracellular matrix. GelMA is a modified form of gelatin that has been double-bonded and solidified by crosslinking under the influence of a photoinitiator through ultraviolet and visible light exposure. That said, the stiff and soft matrix composed of the same material but had varying stiffness due to differences in crosslinking of times by 405 nm light.

2. The in vivo mouse model that carries MYH11-Cre and is tamoxifen sensitive should result in KO of YAP/TAZ in all smooth muscle cells and not just penile SMCs. However, a confounder of that model is that such treatment likely affects resting blood pressure (as vasodilation should increase in multiple vascular beds) and could compromise penile erection due to poor blood flow in general (rather than specifically affecting YAP/TAZ activity in the penile tissue). The authors should determine vasodilatory state in the mouse before and after such systemic injection of tamoxifen and global SMC KO. To rectify, the authors may need to perform the same experiment with direct KO only in penile tissue (i.e., AAV-Cre injected into penile tissue of YAP/TAZ f/f mice).

Response: We are grateful for your suggestions and have taken them into consideration. As per your advice, we locally injected AAV-Cre and Adenovirus-Cre in the penis of *Yap^{f/f};Taz^{f/f}* mice to create penis-specific knockout mice. Our findings indicate that YAP/TAZ was specifically deleted in the penis, as evidenced by the no expression change of YAP/TAZ in the aorta (Fig. S7h). All of the knockout mice exhibited erectile dysfunction (ED), as shown in Fig. 6o-p, columns 5 and 7 in comparison with control. Next, in order to exam the influence of YAP/TAZ SMC's deletion in blood pressure, we checked Yap cKO mice (*Myh11Cre^{ERT2}; Yap^{f/f}*;

Taz^{fl/+}), which had homozygous deletions of YAP and heterozygous deletions of TAZ. Our results revealed that the Yap cKO mice had a significant impact on erectile function, while it had minimal effect on blood pressure (Fig. 6o-p and S7e-f).

3. In that vein, a powerful in vivo experiment that is missing is to use such a penile-specific KO of YAP/TAZ to show that VED treatment is not effective in these mice.

Response: We agree. We concur with the perspective that if our model is accurate, then the YAP/TAZ-ADM pathway should mediate the impact of mechanical stimuli on penile erection. Hence, supplementary mechanical stimuli are unlikely to provide a cure for ED in YAP/TAZ-deficient mice. Our recent findings, presented in Fig. 6o-p, demonstrate that VED does not produce a curative effect in either SMC-specific or penile-specific YAP/TAZ-deficient ED mice (the localized injection of Cre-expressing virus, as you suggested). Furthermore, we have demonstrated that ADM acts as a downstream target of YAP/TAZ, and supplementation of ADM in YAP/TAZ-deficient ED mice can partially restore erectile function in contrasts to no effect with PDE5i treatment (Fig. 5h compare to Fig. 3u). We believe that these findings provide strong evidence to support our proposed mechanism.

4. The authors should comment on whether the use of YAP/TAZ inhibitors that are discussed for cancer may be an issue with erectile dysfunction if they are administered systemically. Since the YAP inhibitor verteporfin is used as a FDA approved drug for ophthalmologic disease, is there any evidence that such patients suffer from erectile dysfunction?

Response: Currently, there is limited information available regarding the relationship between verteporfin and erectile dysfunction, and it is believed that the relationship may be influenced by factors such as dosage and half-life (5-6 hours) (PMID: 12017349 and 10755329). In our study, the dose of verteporfin administered to mice was 100 mg/kg, which equates to approximately 7000 mg in adults. However, it is important to note that the typical dose used for ophthalmologic treatment is only 3 to 20 mg/m², equivalent to 6-40 mg when converted to body weight. There is a substantial difference between the typical human dose and the dose used in our study.

Minor:

-- English and grammar may need to be improved. For example, Fig. 3 title "Mechanically stretches dominate..." does not make sense.

Response: Thanks. We have revised the manuscript.

REVIEWER COMMENTS

Reviewer #1 (Remarks to the Author):

Thanks for the work of editors and authors. Authors have made appropriate revision according to comments. However, I still have some question which may help readers to understand this article easily.

1. line-70. "ED patients who exhibited flaccid/unstretched penis would lose the mechanical stimulation". But the results were base on observation of mice experiment. This article contains both human and mouse result, because the etiology of ED patients is more complex, I suggest that the author make a strict distinction between humans and mice when describing the results to avoid creating dyslexia. The author can integrate the two in the discussion section.

2. fig 1d. How author get the flaccid penis and erectile penis of mice? Whether the penis is removed in an erect state after electrical stimulation. I sincerely invite the authors to further describe the experimental details in the methods section.

3. fig 1m. The activation mechanism of YAP in vivo is complex and may even be influenced by the interaction of other signaling pathways, such as Wnt-Hippo crosstalk. Therefore, it may not be rigorous enough to conclude that a single YAP downstream target gene expression is responsive to mechanical forces. p-FAK and p-MLC may be used to directly reflect the activation of mechanical force transduction signaling pathways.

4. Thank you for the addition of snRNA-seq to address the impact of transient mechanical forces in scRNA-seq. However, the article PMID: 35879305 also used in situ immunofluorescence staining to demonstrate YAP expression and nuclear translation in cavernous tissues of normal and diabetic ED patients. I advise the author not to shy away from discussing this contrast. This may be due to the fact that the effect of diabetic status is not entirely through mechanical force, or early and late stages of disease, which is a point worth further discussion. I do not require the author to give an exact answer in this paper, but it needs to be compared with previous report in the discussion section.

5. We invite the authors to further discuss the effects of species, diseases, and sequencing methods on mechano-Yap signaling, which is essential for subsequent understanding of the mechanisms of exogenous mechano-therapy. For example, in fig1j, GSEA was used to determine whether hippo was activated or inhibited in different cells in addition to enrichment.

6. also the scRNAs-seq, previous articles (PMID: 35879305) have shown that the cavernous sinuses of the penis are heterogeneous with penile arterial and venous, and cavernous sinuses endothelium was more seriously injured than the vascular endothelium in nondiabetic and diabetic ED patients. In the classical theory, endothelial interaction with smooth muscle is the core cause of erection. Therefore, I suggest that the authors follow the standard of previous articles to look at whether different subtypes of endothelial cells and smooth muscle cells differ in the mechanical force signal led by YAP.

7. Similar to the chicken-and-egg problem, the authors suggest that mechanical force disturbances are the cause of many types of ED, however, it seems to be more of an outcome or a node in a vicious cycle in some ED with a clear etiological trigger. At present, the most important clinical concerns are nerve injury ED and diabetes ED after radical resection of prostate cancer. It is obviously not comprehensive to consider mechanical dysfunction as the cause but not the result of these two diseases. I invited the author join in discussion part the contrast, including the expression of some key molecules in this paper such as ADM..., the data of normal, non-diabetic ED and diabetic ED is easy to check it in our recent work on <http://malehealthatlas.cn/>.

All in all, we affirm the efforts made by the authors in this round of revision, which has very important reference significance for our understanding of the etiology and treatment of ED. But since there are many unknown and contrast in the field, it may be misleading to indiscriminately emphasize YAP as a protective factor for ED, and I suggest that the authors address the above issues in the discussion section of the new revision.

Reviewer #2 (Remarks to the Author):

During the 1st revision process, the authors have adequately addressed all issues raised by the reviewer.

Reviewer #3 (Remarks to the Author):

The authors have been responsive to the critiques, and they offer a substantially improved manuscript. I have no additional concerns.

POINT TO POINT TO ANSWER REVIEWER COMMENTS:

Reviewer #1 (Remarks to the Author):

Thanks for the work of editors and authors. Authors have made appropriate revision according to comments. However, I still have some question which may help readers to understand this article easily.

1. line-70. "ED patients who exhibited flaccid/unstretched penis would lose the mechanical stimulation" . But the results were base on observation of mice experiment. This article contains both human and mouse result, because the etiology of ED patients is more complex, I suggest that the author make a strict distinction between humans and mice when describing the results to avoid creating dyslexia. The author can integrate the two in the discussion section.

Response: Thanks for your suggestion. We have revised the manuscript.

2. fig 1d. How author get the flaccid penis and erectile penis of mice? Whether the penis is removed in an erect state after electrical stimulation. I sincerely invite the authors to further describe the experimental details in the methods section.

Response: Thank you for your feedback. We have updated the methods section with a more detailed description of the sample preparation process for AFM measurement in Figure 1d. Specifically, both flaccid and erectile penises were treated similarly to the RNA seq data. After treatment with VED for 15 minutes, the penises were rapidly frozen in liquid nitrogen. The frozen tissue blocks were then cut into 20 mm thick sections for AFM measurement.

3. fig 1m. The activation mechanism of YAP in vivo is complex and may even be influenced by the interaction of other signaling pathways, such as Wnt-Hippo crosstalk. Therefore, it may not be rigorous enough to conclude that a single YAP downstream target gene expression is responsive to mechanical forces. p-FAK and p-MLC may be used to directly reflect the activation of mechanical force transduction signaling pathways.

Response: We respectfully disagree. Our study did not solely conclude that YAP/TAZ response to mechanical forces was based on YAP/TAZ target genes. In fact, our results systematically demonstrate the curative effects of mechanical forces on erectile dysfunction (ED) through the activation of YAP/TAZ. In our in vivo experiments, we applied mechanical treatments such as VED and SWT to mice, which significantly increased YAP/TAZ activity through nuclear localization, higher TAZ expression, and increased expression of YAP/TAZ target genes. Furthermore, as suggested by Rev3, we conducted a rescue experiment by locally depleting YAP/TAZ from the penis, which resulted in the loss of the curative effects of mechanical treatment (as shown in Fig. 6o-p). We also conducted in vitro experiments using various mechanical stimuli, including mechano-related inhibitors, cell stretch, and ECM regulators, all of which resulted in the repression of YAP/TAZ activity at multiple levels (as shown in Fig. 3 and Fig. S3). However, we have followed your suggestions and added more detailed discussion to explain our results further, which we believe will enhance understanding.

4. Thank you for the addition of snRNA-seq to address the impact of transient mechanical

forces in scRNA-seq. However, the article PMID: 35879305 also used in situ immunofluorescence staining to demonstrate YAP expression and nuclear translocation in cavernous tissues of normal and diabetic ED patients. I advise the author not to shy away from discussing this contrast. This may be due to the fact that the effect of diabetic status is not entirely through mechanical force, or early and late stages of disease, which is a point worth further discussion. I do not require the author to give an exact answer in this paper, but it needs to be compared with previous report in the discussion section.

Response: Thanks. We have added the discussion in the manuscript.

5. We invite the authors to further discuss the effects of species, diseases, and sequencing methods on mechano-Yap signaling, which is essential for subsequent understanding of the mechanisms of exogenous mechano-therapy. For example, in fig1j, GSEA was used to determine whether hippo was activated or inhibited in different cells in addition to enrichment.

Response: We agree. We have added the GSEA results of YAP/TAZ target genes in cell clusters and discussion in manuscript.

6. also the scRNAs-seq, previous articles (PMID: 35879305) have shown that the cavernous sinuses of the penis are heterogeneous with penile arterial and venous, and cavernous sinuses endothelium was more seriously injured than the vascular endothelium in nondiabetic and diabetic ED patients. In the classical theory, endothelial interaction with smooth muscle is the core cause of erection. Therefore, I suggest that the authors follow the standard of previous articles to look at whether different subtypes of endothelial cells and smooth muscle cells differ in the mechanical force signal led by YAP.

Response: Thanks. You shared the same option with Rev2. Our single-nucleus RNA sequencing analysis of endothelial cell gene profiles during erection revealed significant changes in the Hippo pathway and YAP/TAZ activity (Fig. S8b). However, we did not observe positive YAP/TAZ immunofluorescence signals in penile sections nor did we obtain supportive data from GSEA. Furthermore, our findings suggest that endothelial cells are not affected by YAP/TAZ or ADM activation (Fig. S8h). These results indicate that YAP/TAZ may respond to mechanical stimuli in endothelial cells, but its effect on endothelial cell function may be limited during the window of detecting restoration of erectile function. Interestingly, we did observe changes in YAP/TAZ activity in pericytes during ED progression. Our results demonstrate a correlation between YAP/TAZ activity in pericytes and ED progression in rats following BCNI treatment. Furthermore, mechanical stimuli treatment was found to significantly enhance YAP/TAZ activity in pericytes compared to non-treated BCNI-ED rats, providing evidence of a mechano-YAP/TAZ cascade in pericytes (Fig. S8e-g). Given the distinct responses of pericytes and endothelial cells, we hypothesize that the mechano-YAP/TAZ pathway may play a role in the vascular system.

7. Similar to the chicken-and-egg problem, the authors suggest that mechanical force disturbances are the cause of many types of ED, however, it seems to be more of an outcome or a node in a vicious cycle in some ED with a clear etiological trigger. At present, the most important clinical concerns are nerve injury ED and diabetes ED after radical resection of prostate cancer. It is obviously not comprehensive to consider mechanical

dysfunction as the cause but not the result of these two diseases. I invited the author join in discussion part the contrast, including the expression of some key molecules in this paper such as ADM..., the data of normal, non-diabetic ED and diabetic ED is easy to check it in our recent work on <http://malehealthatlas.cn/>.

Response: Thank you for sharing your comprehensive work, which provides detailed information on various types of ED. Our study adds to this knowledge by highlighting the crucial role of blood inflow into the corpus cavernosum and resulting mechanical stretching of penile smooth muscle cells (SMCs) in activating the YAP/TAZ-ADM axis, which sustains penile SMC relaxation. While we acknowledge that various factors can cause ED, we emphasize the significance of mechano-YAP/TAZ in erectile function. We found that mechanical-related treatments can effectively cure ED patients, particularly those taking PDE5 inhibitors, even in cases where ED is caused by factors such as nerve injury or diabetes. Our findings can contribute to the development of new therapeutic strategies for ED patients. Thank you again for sharing your work and insights.

All in all, we affirm the efforts made by the authors in this round of revision, which has very important reference significance for our understanding of the etiology and treatment of ED. But since there are many unknown and contrast in the field, it may be misleading to indiscriminately emphasize YAP as a protective factor for ED, and I suggest that the authors address the above issues in the discussion section of the new revision.

Response: Thank you for your affirmation. We have carefully revised the manuscript and incorporated your valuable suggestions, including more detailed discussions, to improve the clarity and impact of our findings.

REVIEWERS' COMMENTS

Reviewer #1 (Remarks to the Author):

Great work, I have no further question. cong!